http://dx.doi.org/10.1098/rsob.19.0258

Subject Area:
biochemistry/bioinformatics/molecular biology

Keywords:
negative stain, haemocyanin, phenoloxidase, myriapods, centipedes, Scolopendra

Authors for correspondence:
P. I. Silva Jr
e-mail: pisjr@butantan.gov.br
R. V. Portugal
e-mail: rodrigo.portugal@lnnano.cnpem.br

†Both authors contributed equally to this work.

# Myriapod haemocyanin: the first three-dimensional reconstruction of *Scolopendra subspinipes* and preliminary structural analysis of *S. viridicornis*

K. C. T. Riciluca[1,2,†], A. C. Borges[1,†], J. F. R. Mello[1], U. C. de Oliveira[2], D. C. Serdan[1], A. Florez-Ariza[1], E. Chaparro[2,3], M. Y. Nishiyama-Jr[2], A. Cassago[1], I. L. M. Junqueira-de-Azevedo[2], M. van Heel[1], P. I. Silva Jr[2,3] and R. V. Portugal[1]

[1]Laboratório Nacional de Nanotecnologia (LNNano), Centro Nacional de Pesquisa em Energia e Materiais (CNPEM), CEP 13083-970, Campinas, Brazil
[2]Laboratório de Toxinologia Aplicada (LETA), Centro de Toxinas, Imuno-Resposta e Sinalização Celular (CeTICS/CEPID) — Instituto Butantan, São Paulo, Brazil
[3]Interunidades em Biotecnologia, Universidade de São Paulo, São Paulo, Brazil

UCdO, 0000-0002-4811-757X; AF-A, 0000-0003-2429-4659; MYN-J, 0000-0002-2410-0562

Haemocyanins (Hcs) are copper-containing, respiratory proteins that occur in the haemolymph of many arthropod species. Here, we report the presence of Hcs in the chilopode Myriapoda, demonstrating that these proteins are more widespread among the Arthropoda than previously thought. The analysis of transcriptome of *S. subspinipes subpinipes* reveals the presence of two distinct subunits of Hc, where the signal peptide is present, and six of prophenoloxidase (PPO), where the signal peptide is absent, in the 75 kDa range. Size exclusion chromatography profiles indicate different quaternary organization for Hc of both species, which was corroborated by TEM analysis: *S. viridicornis* Hc is a $6 \times 6$-mer and *S. subspinipes* Hc is a $3 \times 6$-mer, which resembles the half-structure of the $6 \times 6$-mer but also includes the presence of phenoloxidases, since the $1 \times 6$-mer quaternary organization is commonly associated with hexamers of PPO. Studies with Chelicerata showed that PPO activity are exclusively associated with the Hcs. This study indicates that *Scolopendra* may have different proteins playing oxygen transport (Hc) and PO function, both following the hexameric oligomerization observed in Hcs.

## 1. Introduction

Myriapoda is a group of arthropods divided into four classes, namely Chilopoda (centipedes), Diplopoda (millipedes), Symphyla and Pauropoda [1].

Millipedes are a mega-diverse group of arthropods and are among the most important consumers of detritus in many terrestrial ecosystems. Comprising more than 12 000 described species [2], millipedes are found on six continents and in virtually all of Earth's biomes [3]. This group is characterized by segments in which two pairs of legs are arranged on one body segment [4].

Pauropoda and Symphyla are small, translucent, soil-dwelling myriapods, with body lengths of less than 2 mm and 1–8 mm, respectively. The symphylids have long and filiform antennae and a pair of specialized appendages at the preanal segment, called spinnerets, while the pauropods have distinctive antennae, which are branching and have long flagella. A total of 835 pauropods species in two orders and five families, and 195 symphylid species in one order and two families, have been described to date [5,6].

Centipedes (Chilopoda), one of the four major lineages of myriapods (Arthropoda), constitute the only predatory myriapod group. They live in many terrestrial habitats [7] with fossil records spanning 420 Myr. A key trait of this group is a pair of poisonous claws formed from a modified first appendage [4]. They comprise approximately 3300–3500 species distributed in all continents except Antarctica [8], with the greatest diversity occurring in the tropics and warm temperate zones. Most species inhabit leaf litter and soil or are found under stones, bark or wood in forests, although grassland, desert, caves and the coastal areas are also occupied by some species [9].

The internal organs of all centipedes, except Scutigeromorpha, are supplied with oxygen through trachea, spirally thickened chitinous tubules of ectodermal origin, and which originate from laterally placed openings, the spiracles [10]. Some species have occludable spiracles, a prerequisite for discontinuous respiration and their tracheal ultrastructure is similar to that in insects and arachnids. Other species have tracheal lungs with short tracheal tufts, while still others have an insect-like tracheal system [11].

The tracheal and circulatory systems in Myriapoda are well developed. The circulatory system of chilopods is of the open kind [12]. Analogous to blood, the fluid that circulates in arthropods is called haemolymph. It is pumped by the heart into a body cavity called a haemocoel, where it sloshes around and bathes the internal organs in nutrients and gases [13]. The haemolymph of many arthropods and molluscs presents a protein called haemocyanin (Hc), a large copper-containing protein that transports and stores oxygen [14,15]. Hc is not found in blood cells but is found freely dissolved in the haemolymph. It forms the major protein constituent (50% to 90%) of this fluid, with concentrations of up to $120 \, \text{mg ml}^{-1}$ [16]. Besides Hc, prophenoloxidase (PPO) is also present in the haemocytes as an inactive proenzyme [17]. These proteins are important for primary immune responses, with Hc fragments in plasma acting like antimicrobial peptides (PvHCt [18] and rondonin [19]), and phenoloxidase (PO) enzymes acting as wound healer and playing a role in sclerotization and melanization [20–22].

The melanin is specifically formed from polymerization of dopaquinone, which is generated by the monooxygenation of tyrosine, in turn mediated by the enzyme tyrosinase (TY). Alternatively, dopaquinone or dopamine can also be produced from L-DOPA using a related enzyme, catechol oxidase (CO). TY and CO are commonly referred to as POs [14,23]. TY, CO and PO have similar active sites in which two copper ions, named CuA and CuB, are coordinated by six histidine residues [24,25].

TY belongs to the 'type-3 copper' family along with COs—TY catalyses both the o-hydroxylation of monophenols to o-diphenols and the oxidation of o-diphenols to o-quinones, whereas CO only catalyses the second reaction [21,24,26]. Hc proteins are oxygen carriers from the haemolymph of molluscs and arthropods [27,28].

The basic structure of arthropod Hc is hexameric in nature and is usually found as single or multiple hexamers ($2 \times 6$-mer to $8 \times 6$-mer). Each arthropod Hc subunit (approx. 70–75 kDa) consists of three domains: domain I contains five to six α-helices, domain II contains a four α-helix bundle and domain III is a seven stranded anti-parallel β-barrel. Domain II also encompasses the di-copper centre [14,

28–32]. The geometry and coordination environment of the active site of the arthropod PO are very similar to those of the arthropod Hc; and while not all POs are hexameric [33], those from crustaceans are [34,35].

Arthropod Hc proteins have been best studied in chelicerates and crustaceans [32,36–43]. Hc has also been found in Hexapoda [44–47] and Onycophora [48], but these subphyla, including Myriapoda, have long been ignored in this regard, because of the mediation of their $O_2$ supply via their trachea [49].

Within the myriapods, Hc was first identified in the centipede *Scutigera longicornis* (Chilopoda) [50]. The closely related species *Scutigera coleoptrata* has a $6 \times 6$-mer Hc (36 subunits), which comprises five distinct subunit types, but only four were amenable to analysis using 3D electron microscopy [50–54]. A similar $6 \times 6$-mer Hc is also present in the haemolymph of *Spirostreptus* sp. (Spirostreptida, Diplopoda), although it also contains a significant presence of the $3 \times 6$-mer form [55,56]. The same was found for *Polydesmus angustus* with its $3 \times 6$-mer HC analysed using 3D electron microscopy [57].

Except for the Hc proteins of Scutigeromorpha, Spirostreptida and Polydesmida no information about the 3D structure of any other Myriapoda HC has yet been published, to the best of our knowledge. Here we describe the sequencing and structural analysis of two subunits of Hc and six of PO from the centipedes *Scolopendra subspinipes subspinipes* and *Scolopendra viridicornis*.

# 2. Results

## 2.1. Sequencing of *Scolopendra subspinipes subspinipes* haemocyanin

Pooled haemolymph from six *S. subspinipes subspinipes* organisms were used to construct a library of cDNA fragments for sequencing. The cDNA library was sequenced, resulting in 14 964 551 high-quality filtered reads. The assembly with TRINITY software produced 265 536 contigs. These contigs were compared with Swissprot proteins using Blastx. 72 of the contigs showed similarity to Hc, were annotated as 'haemocyanin' and were curated manually.

The transcriptome analysis of these 72 contigs revealed it to contain seven complete sequences (two isoforms) and only one incomplete sequence (The N-terminus was not sequenced). The analysis suggested the presence of a typical signal peptide of 15 amino acid residues used for transmembrane transport and export into the haemolymph in one sequence, in contrast to the five sequences (two Hc and three POs sequences) identified in *Scolopendra subspinipes dehaani* [58,59]. Two sequences of the above-described seven from *S. subspinipes* have similarity to Hc, with the presence of the signal peptide, and six to PO, characterized by the absence of the signal peptide.

We also analysed the expression of the mRNAs of Hc and PPO subunits using RNA-Seq. We found highly divergent expression levels of Hc subunits and PO, which varied by several orders of magnitude (table 1). To construct the cDNA library, we collected the total haemolymph; it could explain the low level of FPKM (fragments per kilobase of exon per million fragments mapped) of Hc Ssu1 and Ssu2 transcripts because according previous studies, the

**Table 1.** Haemocyanin (Ssu1 and Ssu2) and phenoloxidase (Ssu3–Ssu8) mRNA expression in myriapods. The mRNA levels of Hc and PPO subunits were determined by RNA-Seq based on the transcriptome and displayed as FPKM values.

| contigs | | IP | AA | MW (Da) | MW (Da) (IAA) | FPKM |
|---|---|---|---|---|---|---|
| TRINITY_DN52177_c1_g1_i1 | Ssu1 | 7.21 | 659 | 76 125.25 | 76 477.90 | 0.58 |
| without signal P | | 7.19 | 635 | 73 325.83 | 74 757.94 | |
| TRINITY_DN54115_c0_g1_i1 | Ssu2 | 6.15 | 662 | 76 704.20 | 77 056.33 | 0.81 |
| without signal P | | 6.15 | 643 | 74 748.77 | 75 102.22 | |
| TRINITY_DN54098_c0_g1_i1 | Ssu3 | 6.52 | 657 | 76 938.72 | 77 175.90 | 247.55 |
| TRINITY_DN56304_c0_g1_i1 | Ssu4 | 6.12 | 665 | 76 555.85 | 76 793.62 | 262.59 |
| TRINITY_DN63027_c0_g1_i1 | Ssu5 | 5.28 | 665 | 77 121.87 | 78 213.67 | 27.16 |
| TRINITY_DN63027_c0_g1_i2 | Ssu6 | 5.47 | 661 | 76 670.46 | 77 648.64 | 35.47 |
| TRINITY_DN63588_c1_g1_i2 | Ssu7 | 6.16 | 665 | 76 816.33 | 77 053.67 | 2716.32 |
| TRINITY_DN75346_c2_g10_i7 | Ssu8 | 5.32 | 648 | 75 263.83 | 75 786.68 | 17.23 |

hepatopancreas is the principal site of Hc synthesis in adult crustaceans, hexapods and myriapods [36,56,60]. The highest FPKM value was found for the Ssu7 sequence 2716.32. A Bayesian tree analysis of this sequence (figure 1) suggested that this protein could play a different role in this animal.

## 2.2. Phylogenetic analysis of myriapod haemocyanins and phenoloxidases

The two sequences Ssu1 and Ssu2 (TRINITY_DN52177_c1_g1_i1 and TRINITY_DN54115_c0_g1_i1, respectively) of Hc from *Scolopendra subspinipes subspinipes* were found to be very similar to the two sequences (SMH67860.1 (SdeHcA) and SMH67861.2 (SdeHcB), respectively) of Hc of *S. subspinipes dehaani*, with 91% identity with subunit SdeHcA and 92% identity with subunit SdeHcB. The sequence Ssu1(TRINITY_DN52177_c1_g1_i1), similar to subunit SdeHcA, was found to present a signal peptide of 15 amino acid residues, and the sequence Ssu2, similar to subunit SdeHcB, was observed to be not complete at the N-terminus (TRINITY_DN54115_c0_g1_i1). The six copper-coordinating histidines have been shown to be strictly conserved in all arthropod Hcs [41,61] and present in the copper-binding sites A and B. The amino acid sequence of *S. subspinipes subspinipes* Hc (signal peptide present) and PO (signal peptide absent) was included in a multiple sequence alignment of a total of 62 amino acid sequences of Hc and 17 of PO from arthropods (see electronic supplementary material, figure S1).

From the alignment and Bayesian phylogenetic analysis, two groups with a probable common origin were detected: one group corresponding to the Hcs of Hexapoda (Insecta) and Crustacea, and the other group to the other examples of Hc and PO. Within the second group, the phylogenetic cladrogram indicated three major groups, corresponding to Hcss of Myriapoda and Chelicerata and POs of arthropods. These results were supported by the high-support Bayesian posterior probability of 1.00 (see electronic supplementary material, table S2).

The result of alignment and Bayesian phylogenetic analysis shows the existence of two groups with probable common origin: one corresponds to the Hc of Hexapoda (Insecta) and Crustacea, and another one to the others' Hc and PO. Within the second group, the phylogenetic cladrogram indicated three major groups that correspond to Hc of myriapoda, chelicerata and PO of arthropods. These results are supported by high-support Bayesian posterior probability of 1.00 (electronic supplementary material, table S2).

A previous study of Hcs of myriapods [57] revealed the existence of two distinct clades of Hcs that were separated before the divergence of the myriapod classes—with one of the clades containing subunits of type A and B, and the other containing C and D subunits. In a similar way, in our study, we identified a clade containing the subunits of type A and B found in myriapods, specifically containing: *S. coleoptrata* HcA, HcB and HcX, *P. angustus* Hc1 and Hc3, *Spirostreptus* Hc1, *A. gigas* Hc1, *Scolopendra subspinipes dehaani* HcA, HcB and *S. subspinipes subspinipes* Ssu1 e Ssu2. In the first clade, *S. subspinipes subspinipes* Ssu1 was associated with *S. coleoptrata* HcA and Ssu2 was related to *S. coleoptrata* HcB, and the second clade the *S. coleoptrata* HcC and HcD.

Another study showed the presence of a B-type subunit in all myriapods, suggesting that this subunit may be the central building block of the native Hc [59]. The orthologous C- and D-type subunits occur in both Diplopoda and Chilopoda. A-type subunits appear to be restricted to Chilopoda; the HcX subunit of *S. coleoptrata* is a B-type variant. In Diplopoda, two paralogous HcB subunits are found (HcBI and HcBII). The duplication of the HcB gene was probably the response to the loss of the HcA subunit in this taxon [59]. In this study, we only found the A and B type of HC.

We conducted studies based on the phylogenetic tree and analysis of similarity and the crystallographic hexamer of *Panulirus interruptus* Hc (Protein Data Bank ID 1HCY). The similarity analysis showed that the subunit Ssu1 was more similar to HcA and Ssu2 was similar to HcB, HcC and HcD from *Scutigera coleoptrata*.

The clade formed by POs was subdivided into the crustacean and insect POs and myriapod POs. Notably, the PO of *P. lagurus* (PlaPPO) formed a clade associated with oenychophoran Hc, and *S. maritima* (SmaPPO) was included with *Scolopendra*. Moreover, the myriapod POs were indicated to be more associated with the oenychophoran Hcs than with the pancrustacean POs.

The other six sequences (Ssu3—TRINITY_DN54098_c0_g1_i1; Ssu4—TRINITY_DN56304_c0_g1_i1; Ssu5—TRINITY_DN63027_c0_g1_i1; Ssu6—TRINITY_DN63027_c0_g1_i2;

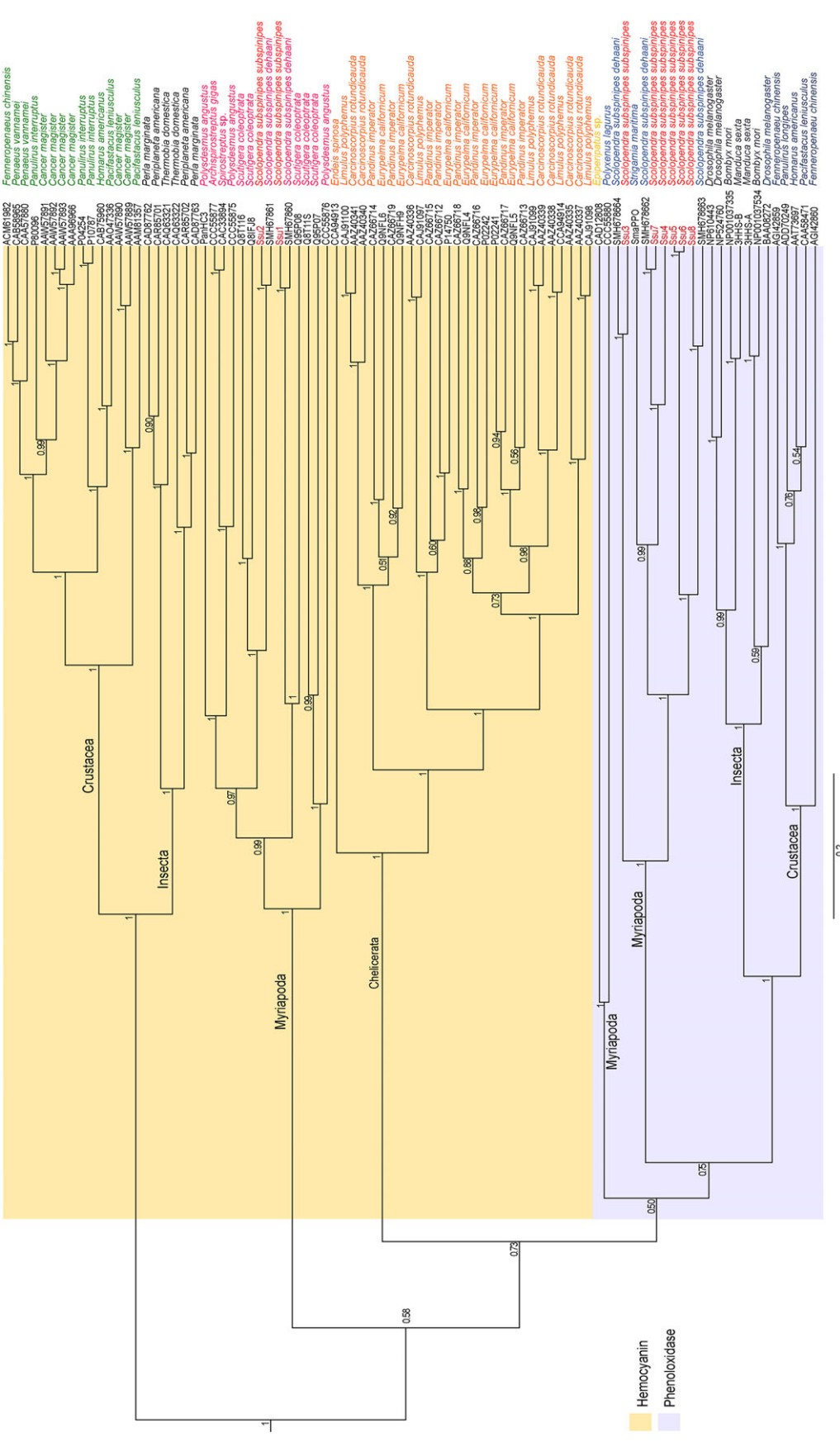

**Figure 1.** Phylogenetic tree of arthropod haemocyanin subunits and phenoloxidase. The numbers at the nodes represent Bayesian posterior probabilities. The species abbreviations are: Myriapoda: Ssu1–Ssu 8—*Scolopendra subspinipes subspinipes*; SMH678663.2, SMH678664.1, SMH678860.1, SMH678861.2—*Scolopendra subspinipes dehaani*; SmaPP0—*Strigamia maritima*; CCC55880.1—*Polyxenus lagurus*; Q95P08.1, Q8IFJ8.1, Q8T11S.1, Q95P07.1, Q8T116.1—*Scutigera coleoptrata*; CAC33894.1—*Spirostreptus sp.*; CCC55877.1—*Archispirostreptus gigas*; CCC55875.1, CCC55876.1, PanHC3—*Polydesmus angustus*; Crustacea: ADD70249.1—*Panulirus longipes*; AAT73697.1—*Homarus americanus*; CAS58471.1, AAM81357.1, AAO47336.1—*Pacifastacus leniusculus*; AGI42859.1, AGI42860.1, ACM61982.1—*Fenneropenaeu chinensis*; CAB75960.1—*Homarus americanu/fcs*; P04254, P10787, P80096.1—*Panulirus interruptus*; AAW57889.1, AAW57890.1, AAW57891.1, AAW57892.1, AAW57893.1, AAA96966.2—*Cancer magister*; CAB85965.1, CAA57880.1—*Penaeus vannamei*; Insecta: BAA08272.1, NP_610443.1, NP_524760.1—*Drosophila melanogaster*; 3HHS-B, 3HHS-A—*Manduca sexta*; NP_001037335.1, NP_001037534.1—*Bombyx mori*; CAQ63321.1, CAQ63322.1—*Thermobia domestica*; CAD87762.1, CAD87763.1—*Perla marginata*; CAR85701.1, CAR85702.1—*Periplaneta americana*; Onychophora: CAD12808.1—*Epiperipatus sp.*; Chelicerata: CCA94913.1—*Endeis spinosa*; CAJ91097.1, CAJ91098.1, CAJ91099.1, CAJ91100.1—*Limulus polyphemus*; AAZ40335.1, AAZ40336.1, AAZ40337.1, AAZ40338.1, AAZ40339.1, AAZ40340.1, AAZ40341.1—*Carcinoscorpius rotundicauda*; CAZ66712.1, CAZ66713.1, CAZ66714.1, CAZ66715.1, CAZ66717.1, CAZ66718.1, CAZ66719.1, CAZ66716.1—*Pandinus imperator*; P14750.3, Q9NFH9.3, Q9NFL6.3, P02242.3, Q9NFL5.3, Q9NFL4.3—*Eurypelma californicum*.

Hemocyanin
Phenoloxidase

royalsocietypublishing.org/journal/rsob   Open Biol. 10: 190258

```
Conservation:                                   555     6  6 95  695   9 896589 558
EcaHcE                      129  KE-AANHP---DQDISVHVVETGN--ILDEEYKLAYFK-EDVGTNAHHWHWHIVYPATWDPAFM----GR  187
EcaHcG                      131  KE-AKNDP---NSDIVVDVQETGN--ILDPEYKLAYFR-EDIGANAHHWYWHVVYPANWDAVFT----GK  189
EcaHcF                      130  KADLKRQSS--DEDVLEIQETGN--ILDPEHKLAYFR-EDIGANAHHWHWHIVYPPTWDASVM----SK  190
EcaHcD_                     130  KVDKVSDP---NKDTVVPIQKTGN--IRDPEYNVAYFR-EDIGINSHHWHWHLVYPAFYDADIF----GK  189
EcaHcA_                     132  TLATTTQPG-DESDIIVDVKDTGN--ILDPEYKLAYFR-EDIGVNAHHWHWHVVYPSTYDPAFF----GK  193
EcaHcC_                     133  KEATRHADK--TDDIIVDMEATGT--IMDPEYNLAYYR-EDIGINAHHWHWHVYPSAWDSVKM----HM  193
EcaHcB_                     132  KEVRLHPDD---EEIIVDIEKTGN--VKDPEYNLAYFR-EDIGVNAHHWHWHLVYPATWRPEVV----HR  191
TRINITY_DN75346_c2_g10_i7  138  KENEKSPDK--PMIPIIIDVEYTST--PLEPEHVLAYWR-EDIGLNVHHWHWHVVYPHEWPH-------NK  196
TRINITY_DN63588_c1_g1_i2   149  TIQRRAPQE-TRETIIIDVDFTGN--DLDPEHRLAYWR-EDIGLNAHHWHWHIVYPYMWIPEL-----GT  209
TRINITY_DN56304_c0_g1_i1   149  TLQRRGEQQ-SQAPIIIDVDFTGT--DLDQEHRLAYWR-EDIALNAHHWHWHIVYPFTWIPEL-----GT  209
TRINITY_DN63027_c0_g1_i1   164  CTGKGLE-----HTRYHVLSCDGN--EPYQEQQLEYWR-QDIWLNAHHWHWHVYPFRVPDT------YQ  219
TRINITY_DN63027_c0_g1_i2   160  CTGKGLE-----HTRYHVLSCDGN--EPYQEQQLEYWR-QDIWLNAHHWHWHVYPFRVPDT------YQ  215
TRINITY_DN54098_c0_g1_i1   147  RHVVLGNGKD-SEREIIIPADFSGN--NLDPEHRLAYWR-EDIQLNSHHWNWHLVYPTDWLPNSG----VG  208
3hhsB                      165  EVSNVVISG-SRMPVNVPINYTAN--TTEPEQRVAYFR-EDIGINLHHWHWHLVYPFDSADRS-----IV  225
3hhsA                      169  EAAAVIPKTIPRTPIIIPRDYTAT--DLEEEHRLAYWRREDLGINLHHWHWHLVYPFSASDEK-----IV  231
3hhs_chainA_p002           169  EAAAVIPKTIPRTPIIIPRDYTAT--DLEEEHRLAYWRREDLGINLHHWHWHLVYPFSASDEK-----IV  231
SsuHcB_TRINITY_DN54115_c0  106  QKALRNDK----DIVVDDTEQHVD--YRDPYSHLGYFL-NDIGLNSHHYHWHVQNSLIWKNRTYPSVVNL  168
SsuHcA_TRINITY_DN52177_c1  138  EEAIRGNQ-----NPVVNLNVSSN--YLNVENRLNYFT-NDLGMNSHHYHYHVVHPATGVPIQ-----VN  194
3wky                       161  KSAIINNQTEVVVEWWHHSDETGLSSRSPEHRVSSYWR-EDMNLNSFHWHWHLSNPYYYIEE------PG  223
3wky_chainA_p003           161  KSAIINNQTEVVVEWWHHSDETGLSSRSPEHRVSSYWR-EDMNLNSFHWHWHLSNPYYYIEE------PG  223
1hcy                       156  SAKMTQ------KPGTFNVSFTGT--KKNREQRVAYFG-EDIGMNIHHVTWHMDFPFWWEDSY-----GY  211
1hcy_chainA_p001           156  SAKMTQ------KPGTFNVSFTGT--KKNREQRVAYFG-EDIGMNIHHVTWHMDFPFWWEDSY-----GY  211
Consensus_aa:                   p.s....p.......hhshp.ots....p.E.pltY@p.pDlshN.HH@HWHls@Ph.h.s.........
Consensus_ss:                   hhhhh          eeeee         hhhhhhhhh hhhhhhhhhhhhh
```

**Figure 2.** Multiple structure alignment of the eight sequences of the myriapod *Scolopendra subspnipes subspinipes*, the seven subunits of spider Hc of *Aphonopelma hentzy* (EcaHc) with the crystal model of Hc of *Panulirus interruptus* (1HCY), PO of crustacean *Penaeus japonicus* (3WKY) and PO of *Manduca sexta* (3HHSA-B) according the secondary structure using the server Promals (http://prodata.swmed.edu/promals/promals.php). The black box showed the amino acid changes that could represent the loss of phenoloxidase activity.

Ssu7—TRINITY_DN63588_c1_g1_i2; Ssu8—TRINITY_DN 75346_c2_g10_i7) of *Scolopendra subspinipes subspinipes* formed a monophyly with the sequences of PO identified in *S. subspinipes dehaani* and *Strigamia maritima* (figure 1).

PO (EC 1.14.18.1) and TY, members of the family of type-3 copper proteins, each catalyses the hydroxylation of monophenol compounds to *o*-diphenol (monophenoloxidase activity) and subsequent oxidation to produce the corresponding *o*-quinone (*o*-diphenoloxidase activity). By contrast, the type-3 dicopper-site-containing members of the protein family called CO catalyse only the latter diphenoloxidase reaction [62,63].

Our inspection of the multiple sequence and structure alignment with seven subunits of spider *Aphonopelma hentzi* (EcaHcA-G), the crystal structure of Hc from the crustacean *Panulirus interruptus* (1HCY) and the crystal structure of PO of *Penaeus japonicas* (3WKY) and of PO of the insect *Manduca sexta* (3HHSA-B) indicated the presence of tryptophane near the first histidine (H1-CuA) in all sequences of PO (3WKY, 3HHSA-B) and sequences of Hc with PO activity (EcaHcA-G); by contrast, our inspection of the sequence/structure of Hc with only oxygen transport activity showed a valine in the crustacean protein (1HCY) and tyrosine in the myriapods ones (figure 2; electronic supplementary material, figure S1) instead of the tryptophan [34,64].

A conserved Phe (F) residue called 'place holder' occurs in the active site pocket before activation and is thought to block access to the substrate present in all type 3 copper proteins. When PPO is activated, this place holder must be removed from the active site pocket. When the blocking residue ($F^{84}$) in DmPPO3 (*Drosophila melanogaster*) was mutated into tryptophan (W), which has a hydrophobic side chain, DmPPO3($W^{84}$) activity significantly decreased after being activated by ethanol [65].

Tryptophan contains a non-carbon atom (nitrogen) in the aromatic ring, making this residue more reactive than phenylalanine although less so than than tyrosine. Tryptophan can play a role in binding to non-protein atoms while the tyrosine side chain, being partially hydrophobic, prefers to be buried in the hydrophobic core. Also, the aromatic tyrosine side chain can be involved in π–π stacking interactions with other aromatic side chains. The valine side chain is small but completely aliphatic and hydrophobic, and hence non-reactive, and is thus rarely directly involved in protein functions like catalysis, although it can play a role in substrate recognition. The hydrophobic aromatic amino acid residues can sometimes substitute for aliphatic residues of a similar size, for example, phenylalanine for leucine, but not the large tryptophan for the small valine. In particular, hydrophobic amino acid residues can be involved in binding/recognition of hydrophobic ligands such as lipids [66].

Although the di-copper active sites of all of the above-mentioned type-3 copper proteins could be well superimposed, interesting differences have been observed, mainly at the CuA site [33]. In particular, tyrosine-switched tryptophan may have made the region less hydrophobic by preventing the entry of substrate, which may explain the probable loss of PO activity from the two Hc proteins identified in *Scolopendra subspinipes subspinipes* (figure 2).

## 2.3. Phenoloxidase activity

In order to test the PO activities of the haemolymphs (lysates of haemocytes and plasma, separately), we performed the microtitre plate assays with lysates of haemocytes and plasma for *Scolopendra subspinipes subspinipes* and *S. viridicornis*. Previous studies showed PO activity in the haemolymph of *S. subspinipes dehaani* [58]. Whereas in insects the proPO may be located in either the blood cells or the plasma, in crustaceans the proPO is mainly located inside the haemocytes [67–69]. Conversely, it has been recently described that PPO from the spiny lobster *Panulirus interruptus* is located in the plasma [70]. Activation of PPO can be induced by limited proteolysis [71] or incubation with components of the blood coagulation cascade, which consists of several serine proteinases [72]. Activation of Hc-derived phenoloxidase-activity *in*

royalsocietypublishing.org/journal/rsob    Open Biol. 10: 190258

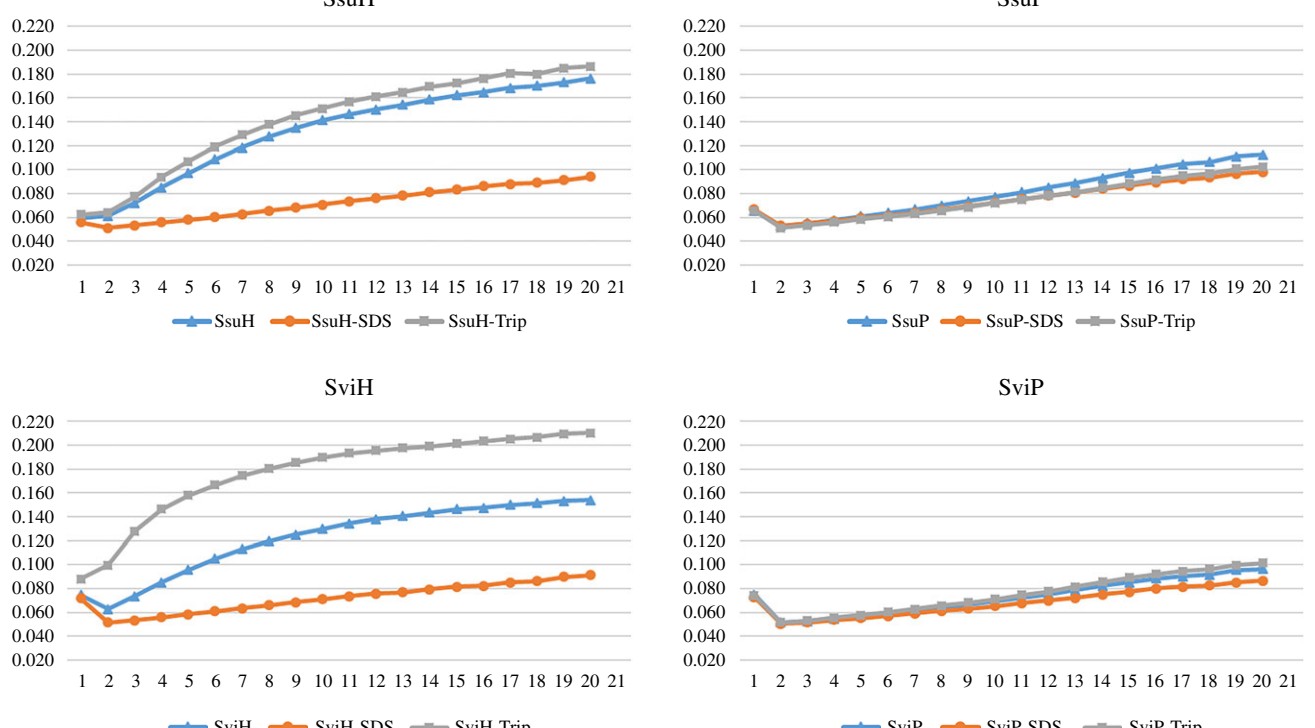

**Figure 3.** Phenoloxidase activity for both species of *Scolopendra*. (*a*) *Scolopendra subspinipes subspinipes* (SsuH—haemocytes, left; SsuP—plasma, right); (*b*) *Scolopendra viridicornis* (SviH—haemocytes, left; SviP—plasma, right). The activity was measured with a Victor³ spectrophotometer (1420 Multilabel Counter/Victor³, Perkin Elmer). The samples (haemocyte (SsuH and SviH) or plasma (SsuP and SviP)) were incubated with dopamine (substrate for PPO) in phosphate buffer with 30 min at 37°C and after that activated with ultrapure water (blue), SDS (orange) or tripsine (grey). The increase of absorbance reveals the phenoloxidase activity by the formation of dopaminechrome.

*vitro* is commonly induced by incubation with the anionic detergent sodium-dodecylsulfate (SDS) [73,74]. However, the activity could also be induced by the presence of free fatty acids and phospholipids [73,75].

In this study, we used a concentration of 20 µg ml⁻¹ from lysate of haemocytes and plasma for both species to verify the *o*-diphenoloxidase incubating with dopamine. The activation of PPO to phenoloxidase by trypsin (square grey) and the activity of the lysate of haemocytes with buffer (triangle blue) occurred for both species (figure 3*a,b*), suggesting that serine protease, the enzyme responsible for this activity, was also present in the haemocytes. Previous studies showed an activation of phenoloxidase activity for myriapods by zymosan and chymotrypsin [76], corroborating our results.

## 2.4. Protein concentration and pH variation

Studies with the spider *Aphonopelma hentzi* (*Eurypelma californicum*) [77] showed that dilution of samples of Hc down to 0.04 mg ml⁻¹ did not cause significant dissociation of Hc complexes. However, association was mainly favoured at concentrations of 0.5 and 1.2 mg ml⁻¹. In our current study, we observed that reassociation of the complexes did not occur easily at concentrations below 0.2 mg ml⁻¹. Thus, for the oligomerization studies, the haemolymph of *Scolopendra subspinipes* was incubated with at 2.6 mg ml⁻¹ Hc and only diluted for the preparation of the electron microscopy grids.

The optimal pH for complex formation and stability were also investigated, since pH changes may cause dissociation. As shown in figure 4, there was a relatively high ratio of the number of 3 × 6-mer complexes to the number of single

hexamers for pH values between 6.2 and 6.6. When the pH value was increased to 7.2–7.8, more single hexamers and fewer 3 × 6-mer complexes were observed. But this tendency reversed course when the pH was increased further to between 7.8 and 8.2, where more 3 × 6-mer complexes were observed.

To verify these proportions, specifically the ratio of the number of 3 × 6-mer complexes to the number of single hexamers, at the various pH levels, we counted the particles (figure 5*a,b*). The results of this counting demonstrated that this sample showed an elevated degree of dissociation in all pH tested. We did not ignore the possibility that a small amount of phenoloxidase was present in the haemolymph used to perform this experiment.

Previous studies with Hcs of arthropods and molluscs mostly worked with buffers having pH values in the range 7.4–8.0, and always in the presence of bivalent calcium and magnesium cations at physiological concentrations [36,37,54,78]. However, the optimum buffer pH varies from species to species, and our search for the optimal pH was pertinent for studies depending on a maximum preservation of the structure of the protein complexes. Thus, according to the results of the current study, the optimal pH for storage of *Scolopendra subspinipes subspinipes* Hc was determined to be in the range 8.0–8.2. The same range of pH was used for *S. viridicornis*.

## 2.5. Structural analysis of *Scolopendra subspinipes subspinipes* haemocyanin

The Hcs of the animals were purified using a Superose 6 3.2/300 (GE Healthcare) SEC column on an FPLC (AKTA) with

royalsocietypublishing.org/journal/rsob Open Biol. 10: 190258

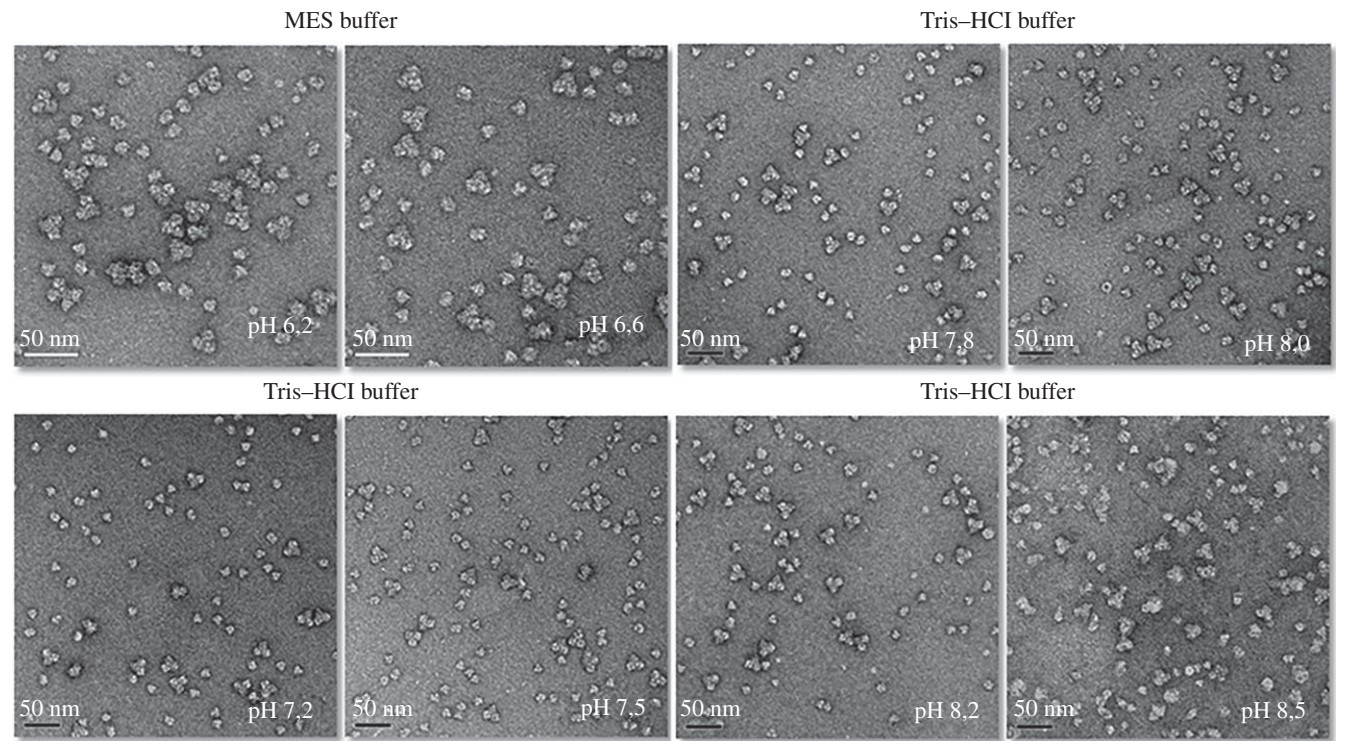

MES buffer · Tris–HCl buffer · Tris–HCl buffer · Tris–HCl buffer

pH 6.2 · pH 6.6 · pH 7.8 · pH 8.0 · pH 7.2 · pH 7.5 · pH 8.2 · pH 8.5

**Figure 4.** pH variation of the buffer solution influences the reassociation of the 3 × 6-mer quaternary complex of myriapod haemocyanin *Scolopendra subspinipes subspinipes*. Haemocyanin samples were incubated in MES and Tris–HCl buffer solutions, pH ranging from 6.2 to 8.5. Samples were contrasted with 2% uranyl acetate. Images obtained in transmission electron microscopy Jeol-1400PLus.

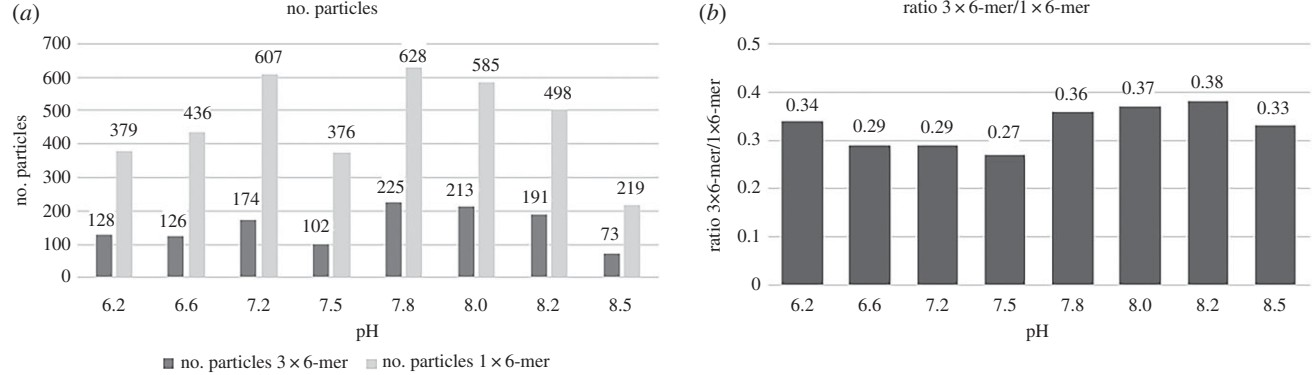

**Figure 5.** The number of particles and ratio between the hexamer and complex at the range of pH 6.2–8.50. (*a*) The number of particles was counted using the program 'Cell Counter' plugin of Fiji software (ImageJ 1.49 V). The 1 × 6-mer was represent by light grey and the complex 3 × 6-mer by grey. (*b*) The ratio of 1 × 6-mer to 3 × 6-mer was calculated.

Tris as stabilizing buffer at a flow of 0.1 ml min$^{-1}$ (figure 6). We found two major peaks in both chromatographic profiles. The first peak corresponded to a high level of oligomerization for the complex. We were initially expecting to find a 6 × 6-mer-type complex of *Scolopendra subspinipes subspinipes* Hc, as was found for Hc from the centipede *Scutigera coleoptrata*, an organism also from the Chilopoda class [54]—but the retention volume of our chromatographic profile (figure 6*a*) indicated only the 3 × 6-mer-type complex. This chromatographic result was corroborated by an analysis of transmission electron microscopy images of negatively stained samples, which showed a 3 × 6-mer-type complex structure, specifically with the three hexamers related to each other by a symmetry rotation axis of 120° and showing a diameter of approximately 29 nm (figure 7).

By contrast, the retention volume of the chromatographic profile (figure 6*b*) of *Scolopendra viridicornis* Hc (figure 8)

indicated the same 6 × 6-mer-type complex as found for *Scutigera coleoptrata* Hc. Note that we were only able to obtain these results when we used the anticoagulant propranolol, which prevents the degranulation of haemocytes. Also note that the *Scolopendra viridicornis* centipede used was found at Santa Barbara do Oeste (a dry region), while *S. subspinipes subspinipes* inhabits the coastal area of Rio de Janeiro.

The 3 × 6-mer structure with D3 symmetry was also observed for the Hc from the diplopod *Polydesmus angustus*. Densitometry results showed the presence of the three subunit types in equimolar proportions, with the 18 subunits forming the complex being evenly distributed among two zones: a core zone with six subunits (PanHc1) and a mantle zone with 12 subunits (PanHc2 and PanHc3) [57].

Since specific interactions have been shown to be important to the assembly of the final 6 × 6-mer structure of

royalsocietypublishing.org/journal/rsob Open Biol. **10**: 190258

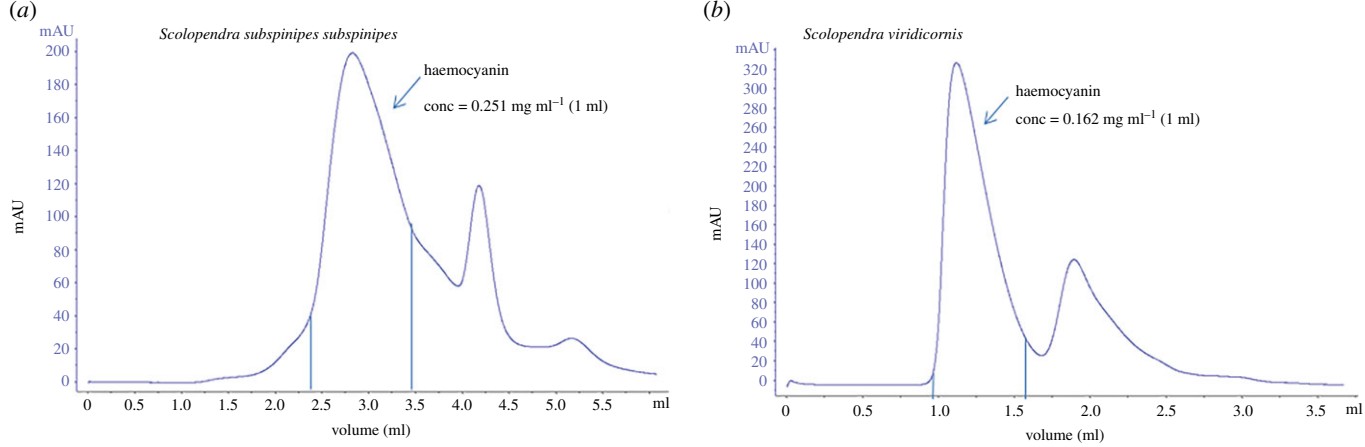

**Figure 6.** Chromatogram profile of plasma. The plasma was separated from haemolymph by centrifugation at 3300*g*, 30 min at 4°C and the supernatant was submitted to purifier AKTA with column Superose 6 3.2/300 (GE Healthcare) with isocratic buffer (Tris Buffer pH 8.2) with flow 0.1 ml min$^{-1}$. (*a*) *Scolopendra subspinipes subspinipes* and (*b*) *S. viridicornis*.

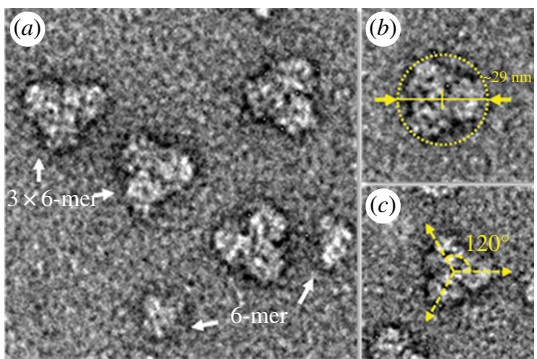

**Figure 7.** Quaternary complex of myriapods haemocyanin *S. subspinipes subspinipes*. (*a*) Haemocyanin samples stained with 2% uranyl acetate were analysed by a Jeol 1400Plus transmission electron microscope. (*b*) The structure found at *S. subspinipes subspinipes* has a quaternary complex consisting of three hexameric subunits and having a diameter of about 29 nm. (*c*) The three hexamers connect at angles of approximately 120° between their axes. It is also possible to observe dissociated hexamers (6-mer).

**Figure 8.** Quaternary complex of myriapode haemocyanin *S. viridicornis*. Haemocyanin samples stained with 2% uranyl acetate. They were analysed by a Jeol-1400Plus transmission electron microscope. (*a*) The structure found at *S. viridicornis* has a quaternary complex consisting of six hexameric subunits. (*b*) The 6 × 6-mer complex. (*c*) Complex of three hexamers was also found.

*Scutigera coleoptrata* Hc, we performed an alignment of its sequence with our two sequences of Ssu1 and Ssu2 (table 2).

Observations of the mantle zone indicated a lack of any contact between two subunits from each hexamer and any other hexamer, and indicated the remaining two subunits to be attached to their counterpart in the adjacent hexamer, together forming the three peripheral bridges. In the 6 × 6-mer assembly of *S. coleoptrata* Hc, this type of bridge has been designated as a 3 ↔ 4 interface [54,57]. This interface in Ssu1 formed from residues 402HYKLTYPA409, 633TH634 and in Ssu 2 from 361VEQLTWPD368, 587TH588.

In the *S. coleoptrata* 6 × 6-mer structure, the two 3 × 6-mer half-structures have been previously noted to be connected in the mantle by a type of inter-hexamer bridge that has been termed a 3 ↔ 3 interface [54]. We observed the presence of histidine (147NRGE**H**YDRIPi156, 442TK**H**444) in the ScoHcC subunit; histidine is an important amino acid able to participate in π–π stacking interactions and it was responsible for stabilizing the structure. Our inspection of the Ssu1 sequence at 141IRGNQ–NPVVNL154 and 432TRQ434 showed a gap in this region and the absence of an amino acid residue able to perform the same interaction. And

our inspection of Ssu2 at 106QKAL**R**NDKDIVVD118 and 401TGP403 showed an arginine in the place of the histidine; the arginine here may support allostery rather than blocking potential dimerization of 3 × 6-mers [54].

The second type of bridge connecting the 3 × 6-mers in the core of *S. coleoptrata* is termed the 1A ↔ 1E interface, with this interface established by interactions of basic residues (KAKK) in the β3D → β3E loop with acidic residues (ED) at the end of helix α3.5. Comparable charges (ENKK and PSD) were observed in this region of Ssu2 (electronic supplementary material, figure S2).

The three central inter-hexamer bridges of the 3 × 6-mer have been observed to be formed by the core subunit Ssu2, and this type of contact has been termed a 1 ↔ 2 interface [54]. Each of the three bridges was formed by a pair of such interfaces together being made up of a large number of residues (table 2). In *S. coleoptrata* Hc, each 1 ↔ 2 interface pair included a histidine cluster that may support allostery [54].

According to our results, the interfaces 3 ↔ 3 and 3 ↔ 4 in *Scolopendra subspinipes subspinipes* showed some differences that corresponded with electron microscope images and its inability to form the 6 × 6-mer assembly.

**Table 2.** Amino acid residues in the region of the four types of inter-hexamers interface from *Scutigera coleoptrata*. These four regions were important to form the complex 6 × 6-mer.

| | |
|---|---|
| mantle interface 3 ↔ 3 | |
| ScoHcC: | 147NRGE**H**YDRIPi156, 442TK**H**444 |
| Ssu1: | 141IRGNQ—NPVVNL154, 432TRQ434 |
| Ssu2: | 106QKAL**R**NDKDIVVD118, 401TGP403 |
| mantle interface 3 ↔ 4 | |
| ScoHcB: | 394SEDLTYND401, 622TK623 |
| ScoHcC: | 397TEELTPQN40 4,627TK628 |
| ScoHcD: | 396KDELNFPD403, 625TP626 |
| Ssu1: | 402HYKLTYPA409, 633TH634 |
| Ssu2: | 361VEQLTWPD368, 587TH588 |
| core interface 1A ↔ 1D | |
| ScoHcA: | 399LNDVTVKP**H**KGD-YDDEV**H**T417 |
| Ssu1: | 412VKEVKVVP**D**KRPEILNEV**R**T431 |
| Ssu2: | 371VEGLTVEG**A**K - - - VNKI**K**T386 |
| core interface 1 ↔ 2 | |
| ScoHcA: | 50PQ**H**EIF55, 89LR**H**RIN94, 143**H**GEEKP148 |
| Ssu1: | 49PKNEIF54, 88LITRIN93, 142**R**GNQNP147 |
| Ssu2: | 17PR**R**EVY22, 56LS**H**RIN61, 110**R**NDKDI115 |

## 2.6. Single particle analysis

Single particle analysis (SPA) was performed using IMAGIC-4D software, following the methodology described previously [79,80]. From the beginning of our analysis, this SPA was conducted using a free-reference particle picking and 2D classification. Later D3 symmetry was imposed during the angular assignment and three-dimensional reconstruction rounds. This D3 symmetry was imposed according to previous structural analysis of Hc from *Polydesmus angustus* [57].

The final three-dimensional reconstruction was obtained from 5998 particles classified into 247 class averages. A Fourier shell correlation (FSC) analysis of the EM reconstructions indicated an estimated resolution of 17 Å using the 1/2-bit threshold criterion (figure 9b). The 3D density map was finally low-pass filtered at 17 Å and one characteristic view is shown in figure 9a. Also, some representative 2D class averages and their respective re-projections are shown in figure 9c.

Three-dimensional models of Ssu1 and Ssu2 were built using the known structure of oxygenated Hc (PDB ID: 1OXY) from *Limulus polyphemus* [28] since they showed 41.01% identity and 40.03% sequence identities, respectively, with all of the histidine residues conserved. The homology models of the subunits are shown in figure 10. Each subunit was constructed separately using SWISS-MODEL (https://swissmodel.expasy.org/).

A multiple sequence and structure alignment with 1OXY showed a gap (of 20 amino acid residues) in the N-terminal region of Ssu1 (electronic supplementary material, figure S4), which explained the loop created at this region (figure 10a) when compared with Ssu2 (figure 10b). In *Polydesmus angustus* Hc, an unusually large histidine-rich loop of 30 amino acid residues (11 histidines) was observed to be located between β-strands 3A and 3B of the subunit PanHc1 [57].

The acquired Ramachandran plot of the Ssu1 model obtained with SWISS-MODEL showed 94.7% of its residues (584 residues) in geometrically favoured regions, 4.5% of the residues (28 residues) in allowed regions and 0.8% of the residues (5 residues) in outlier regions; that of the Ssu2 model obtained with SWISS-MODEL showed 93.2% of its residues (546 residues) in favoured regions, 6.5% of the residues (38 residues) in allowed regions and 0.3% of the residues (2 residues) in outlier regions (http://mordred.bioc.cam.ac.uk/~rapper/rampage.php). Verify3D results showed 81.74% and 87.76% of the amino acid residues in the SWISS-MODEL models having compatible 3D-1D scores greater than 0.2 for Ssu1 and Ssu2 (http://servicesn.mbi.ucla.edu/SAVES/), respectively, suggesting an overall self-consistency for the models in terms of sequence-structure compatibility. These sequences were used to construct the putative hexamer.

Since there is little published information about the chain content of the *S. subspinipes subspinipes* Hc structure, and no published description of how these chains organize to form the hexamer and then the complete 3 × 6-mer oligomer, we set out in the current work to understand how the hexamer is organized. To accomplish this goal, we performed three molecular dockings, using chains Ssu1/Ssu1, Ssu1/Ssu 2 and Ssu2/Ssu 2. The results were compared with *Panulirus interruptus* Hc, a homohexamer organized as 3 tight dimers [29]. The top-ranked solutions of the three dockings were analysed and, interestingly, the Ssu1/Ssu2 interaction profile was found to be very similar to that of the *P. interruptus* tight dimer. This result suggested that *S. subspinipes subspinipes* Hc is probably also organized as three tight dimers (figure 11).

Additionally, the molecular docking results indicated a more dispersive pattern of hotspots for the Ssu2/Ssu2 solutions than for Ssu1/Ssu1 and Ssu1/Ssu2 (figure 12a–c). The top 20% of the docking solution of Ssu1/Ssu1, Ssu1/Ssu2 and Ssu2/Ssu2 are represented by coloured spheres, with dark blue representing a poor score and red a good score, in figure 12d–f. This pattern demonstrated the potential of an Ssu2 chain to interact with another Ssu2 chain in several different ways, or that Ssu2 could be interacting with more than one chain simultaneously. Assuming the last hypothesis, the Ssu2 chain would be localized in the middle (core zone) of the 3 × 6 hexamer and interacting with other Ssu2 chains inside the hexamer (1 ↔ 2 interface), and hence could provide inter-hexamer interactions.

Based on the results from checking the alignment, analysing the regions of interhexamer interactions of *S. coleoptrata* and performing the molecular docking of the two subunits from Hc, the putative hexamer of *S. subspinipes subspinipes* was modelled as consisting of three subunits of Ssu1 and three of Ssu2 arranged in two homotrimers (3 × Ssu1 and 3 × Ssu2).

The Ramachandran plot of the hexamer model obtained with an alignment in PYMOL showed 89.6% of its residues (2906 residues) in the most-favoured regions, 9.2% of the residues (298 residues) in additionally allowed regions, 0.9% of the residues (30 residues) in generously allowed regions and 0.3% of the residues (9 residues) in disallowed regions. Verify 3D results showed 88.35% of the residues in the hexamer model having compatible 3D-1D scores greater than 0.2 (http://servicesn.mbi.ucla.edu/SAVES/).

Based on symmetry considerations previously determined for the *S. coleoptrata* 6 × 6-mer, and applied to the 3 × 6-mer

**Figure 9.** 3D reconstruction of *Scolopendrra subspinipes subspinipes* haemocyanin. (*a*) 3D density map filtered at 17 Å. (*b*) The Fourier shell correlation (FSC) of the EM reconstructions shows an estimated resolution of 17 Å using the 1/2-bit threshold criterion. (*c*) Some representative 2D class averages and their respective re-projections using IMAGIC 4D.

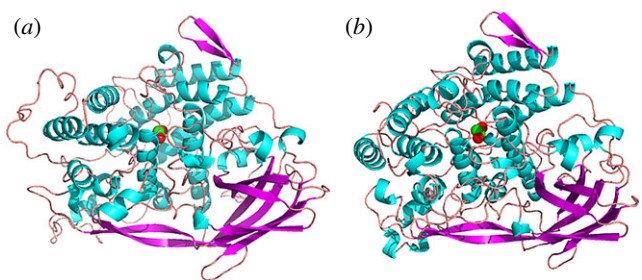

**Figure 10.** Molecular modelling of (*a*) Ssu1 and (*b*) Ssu2 using SWISS-MODEL with *Limulus polyphemus* subunit II (PDB ID: 1OXY) as a template (helix, cyan; sheet, magenta; loop, pink).

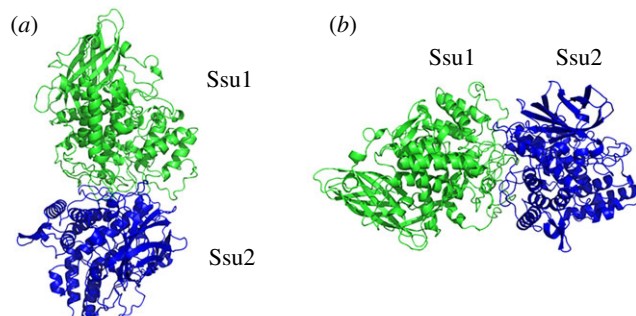

**Figure 11.** (*a*) Molecular docking between subunit Ssu1 and Ssu2 (tight dimer) of *Scolopendra subspinipes subspinipes*. (*b*) Ribbon structure in a different view related by approximately 90° rotation.

found in *S. subspinipes subspinipes*, we suggested that the constituent $1 \times 6$-mers are identical with respect to their subunit composition [54,57]. Considering this hypothesis, we constructed a hexamer using the tight Ssu1/Ssu2 dimer and performed a rigid body fitting of it into the negative stain map using CHIMERA. The map was divided into three regions, corresponding to each hexamer. Three Ssu1/Ssu2 dimers were positioned in the hexamer map so that the Ssu2 subunit could have more intra- and inter-hexamer contact points. Then, each hexamer was fitted individually into the map. The optimal fit was achieved with a correlation of at least 0.6 between the negative stain map and the atomic-model-simulated map (figure 13).

## 3. Discussion

Hcs and related proteins of several arthropod species have been intensively studied, but their presence in Myriapoda has been largely ignored as these organisms had been assumed to not have Hc or any other respiratory protein. Respiratory proteins had been considered unnecessary in this taxon because Chilopoda (centipedes) and Diplopoda (millipedes) possess, similar to the insects, a typical tracheal system that was thought to be sufficient to supply the internal organs with an adequate amount of oxygen [49,81].

The first biochemical evidence for a myriapod Hc was found in the centipedes *Scutigera longicornis* and *S. coleoptrata* [50,51]. More recent data have suggested that Hc is also present in other myriapods, including in at least three of the four myriapod classes, namely Chilopoda, Diplopoda and Symphyla. [55–57,59,82].

Few data are still available on POs of myriapods. To date, only a few biochemical results have demonstrated its presence [83,84]. Previous studies showed indications for an mRNA encoding a phenoloxidase in Diplopoda *Polyxenus lagurus*. In their database analysis, they also did identify a gene for a putative PPO in the fully sequenced genome of the Chilopoda *Strigamia maritima*. Both proteins lack a signal peptide, suggesting them to be intracellular proteins. In addition, in neither species was Hc detected [57].

In the kuruma prawn *Marsupenaeus japonicus*, PPO and Hc are synthesized in the hepatroprancreas with an N-terminal signal peptide [85], and are secreted to plasma [35], whereas the other well-known arthropod phenoloxidase is synthesized in haemocyte cells without the signal peptide [86,87]. Moreover, the quaternary structure of crustacean PO forms a hexamer, similar to that of Hc, whereas the insect PO usually forms a dimer [33,88,89].

In our current work, we identified six sequences of PO from *S. subspinipes subspinipes*. Our phylogenetic analysis put this protein closer to POs of crustaceans and Hexapoda, and to Hcs of Chelicerada, than to Hcs of Myriapoda. The Hcs of Myriapoda were found to be more similar to Hcs from Hexapoda and crustaceans than to Chelicerada Hc. We speculated that the same site of biosynthesis occurs for myriapod Hcs according the analysis of FPKM of mRNA. For this analysis, an extraction was carried out using haemolymphs that could contain mRNA from haemocytes and other organs. The PO activity was found only in lysate of haemocytes for both species of Scolopendra, corroborating the results of FPKM. However, plasma PPO contamination was

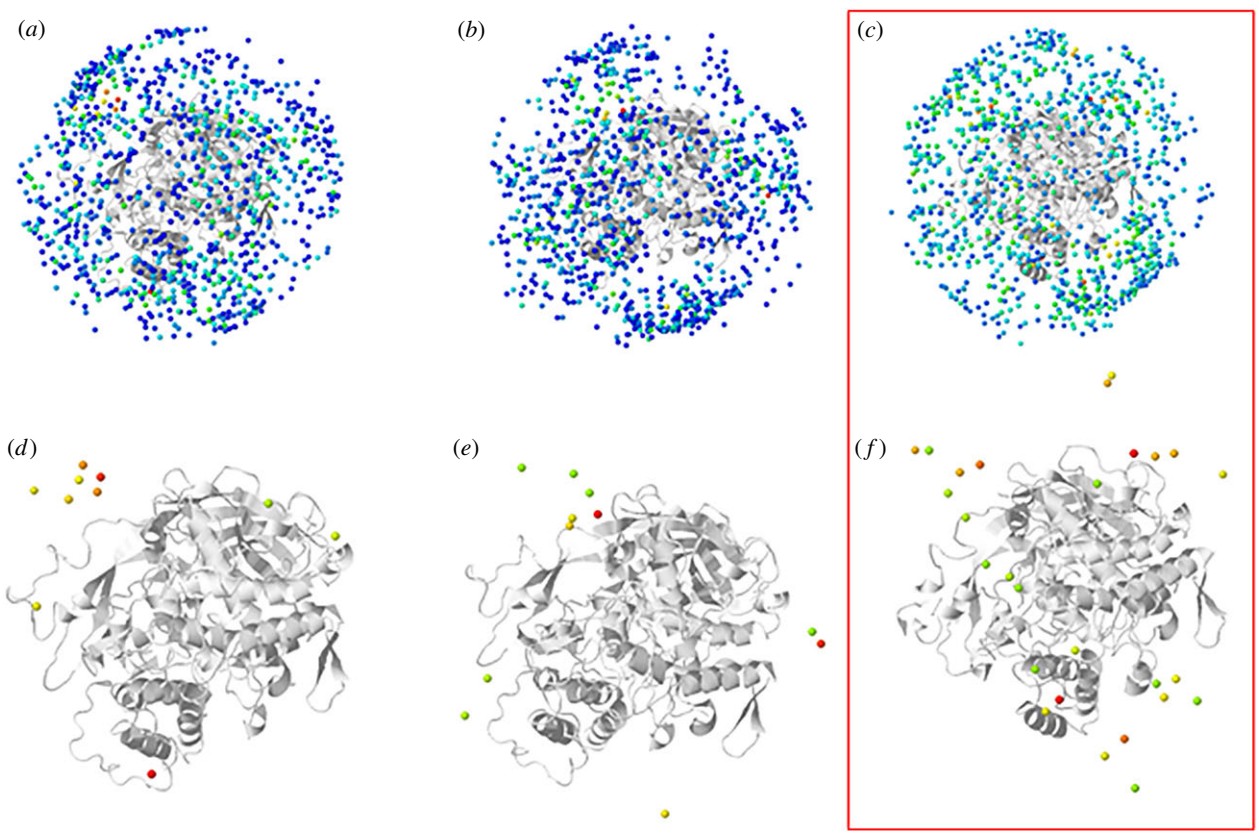

**Figure 12.** Molecular docking hotspots from dark blue (poor score) to red (good score) made using the Frodock server. (a) 100% of the Ssu1/Ssu1 docking solutions, (b) 100% of the Ssu1/Ssu2 docking solutions, (c) 100% of the Ssu2/Ssu2 docking solutions, (d) the best 20% of the Ssu1/Ssu1 docking solutions, (e) the best 20% of the Ssu1/Ssu2 docking solutions, (f) the best 20% of the Ssu2/Ssu2 docking solutions. A more dispersive pattern is present in Ssu2/Ssu2 docking solutions, which can be related to the Ssu2 chain being involved in interactions with other chains providing the inter-hexamer interaction.

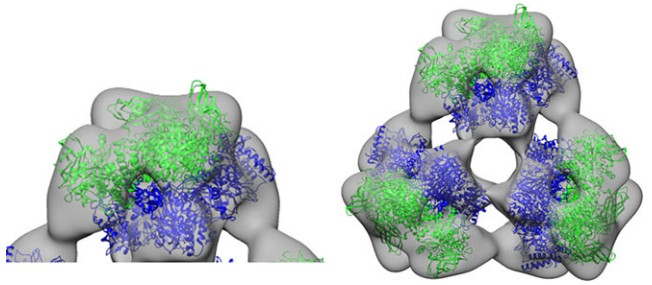

**Figure 13.** Rigid body fitting of 3 dimers generating the hexameric structure. Each hexamer was fitted individually into the map generating the 3 × 6 oligomer of haemocyanin.

ruled out because anticoagulant buffers were used in this study.

The electron microscopy analyses revealed the presence of a 3 × 6-mer structure, the same as that found for the diplopod *Polydesmus angustus* [57]. The evolution of trachea was not necessarily accompanied by a loss of specific respiratory proteins. The 6 × 6-mer structure was found in Chilopoda *Scolopendra viridicorinis* just as in *Scutigera coleoptrata* and was suggested in the *Spirostreptus* sp and *Archispirostreptus gigas* [82]. Whereas the respiratory proteins *Spirostreptus* organisms show high oxygen affinity and low cooperativity, those of *S. coleoptrata show* low oxygen affinity and high cooperativity. The high oxygen affinity for Diplopoda may be related to the low oxygen conditions of the subterrestrial environment, where these species remain most of the time during daylight hours [55,82]. By contrast, Scutigeromorpha organisms are very fast animals that may require

bursts of oxygen in active phases and present short tracheal systems [53,90].

According to studies with *S. subspinipes dehaani*, the affinity for oxygen was indicated to be high but with low cooperativity [58,59]. This result would appear to be consistent with the observation of the 3 × 6-mer structure in the electron microscopy images from *S. subspinipes subspinipes*. Members of this species are fast, like those of *S. coleoptrata*, but they live at moderately hypoxic environments of the subterrestrial habitats in which the animals burrow during the daytime. The $O_2$-binding properties of the Hc are compatible with the role of the circulating haemolymphs in complementing the tracheal system in supplying $O_2$ to the respiring tissues in myriapods [82].

The 6 × 6-mer Hc of the Chilopoda *Scutigera coleoptrata* remains stable in near-neutral buffers and in the presence of $Ca^{++}$ and $Mg^{++}$ ions [54]. By contrast, *Spirotreptus* sp. Hc shows a mixture of 6 × 6-mer and 3 × 6-mer assemblies under such conditions [55,56]. The 3 × 6-mer structure has been indicated to be the major quaternary structure of *Polydesmus angustus* Hc, but the presence of a relatively small amount of a 6 × 6-mer structure in the haemolymph was not ruled out [57]. In our current study, we verified that the best buffer to maintain the stability of the protein was that with a pH in the range of 8.0–8.2 and containing $Ca_2^{++}$ and $Mg_2^{++}$ ions.

While we found a relatively small amount of the 3 × 6-mer Hc form in *Scolopendra subspinipes subspinipes* and found the 6 × 6-mer form in *Scolopendra viridicornis*, phenoloxidases were also found to be present in the haemolymph. Note in this regard that the 1 × 6-mer quaternary organization has

also been shown for hexamers of PPO [34]. More studies of Hc of *Scolopendra viridicornis* will be necessary to explain the presence of 6×6-mer Hc.

The currently achieved resolution of 17 Å for *S. subspinipes subspinipes* Hc did not allow for the construction of a pseudo-atomic model of the 3×6-mer quaternary structure by fitting individual subunits, but we used the crystallographic 1×6-mer of *Panulirus interruptus* Hc to check the possible interactions and to construct tight dimers of *S. subspinipes subspinipes* by molecular docking and fitting into the structure.

The results of our alignment analysis showed a similarity between Ssu1 and ScoHcA (53.28%) and between Ssu2 and ScoHcB (54.92%) of *S. coleoptrata* and *Scolopendra subspinipes dehaani* [54,59]. Therefore, our studies with molecular docking suggested the Ssu2 unit (which showed more hotspots for forming anchoring interactions) to be present in the core zone, and Ssu1 to be a constituent of the mantle zone. This result suggested a different organization than that suggested for the corresponding *Scutigera coleoptrata* and *Polydesmus angustus* proteins [54,57]. According to previous studies, a B-type subunit was found in all myriapod species investigated, suggesting that this polypeptide may be the central building block of the native Hc [59].

The hexamer, a basic structure, has been shown to be conserved in all Hcs. But the 1×6-mer organization was probably the most conserved structure of the last common ancestor of arthropod Hcs, leading to the possibility of different numbers and arrangements of the hexamers for crustacean, chelicerae and myriapods Hcs [38,60] and to the formation of these multi-hexamers having occurred independently in these subphyla. This oligomerization is most likely correlated with an enhanced oxygen transport capacity with less osmotic impact. While among the Crustacea and Chelicerata species, the structures of these multi-hexamers vary, ranging from 1×6 to 8×6 subunits, the 36-mer Hc appears to be unique to the myriapod Hcs. This structure was apparently conserved since the separation of the Diplopoda and Chilopoda, at least 400 Ma if not much earlier [91–93]. The myriapod Hc is a unique structure and can be considered as an additional synapomorphy that favours a monophyly of Myriapoda [56].

# 4. Material and methods

## 4.1. Animals

The myriapods *Scolopendra subspinipes subspinipes* and *Scolopendra viridicornis* (order Scolopendromorpha) were collected at Rio de Janeiro and Santa Barbara D'Oeste, respectively and they were kept alive in the biotherium of the Special Laboratory of Toxinology Applied of the Institute Butantan (São Paulo, Brazil). These animals were collected under License Permanent Zoological Material no. 11024-3-IBAMA and Special Authorization for Access to Genetic Patrimony nos 001/2008 and 010345/2014-0.

## 4.2. RNA isolation, cDNA library construction and sequencing

The haemolymph (approx. 800 µl) from six *Scolopendra subspinipes subspinipes* of both sexes at different stages of development was collected by cardiac puncture with an apyrogenic syringe. To avoid haemocyte degranulation and coagulation, the haemolymph was collected in the presence of sodium citrate buffer (0.14 M NaCl, 0.1 M glucose, 30 mM trisodium citrate, 26 mM citric acid, 10 mM EDTA, pH 4.6 (2:1, v/v)) [94] and pooled in Eppendorf tubes and immediately frozen at −80°C. Total RNA were extracted using TRIzol reagent (Ambion, Life Technologies) according to manufacture protocol. RNA integrity was assessed using an Agilent 2100 Bioanalyzer with the RNA 6000 Nano assay and quantified by Quant-iT RiboGreen RNA reagent and Kit (Invitrogen, Life Technologies Corp.). cDNA library was generated following the standard TruSeq Stranded RNA Sample Prep Kit protocol (Illumina, San Diego, CA, USA). Briefly, purification of the poly-A tail containing mRNAs by ligated magnetic beads poly-T oligonucleotides, fragmentation, synthesis of the first cDNA strand, synthesis of the complementary strand, 3′ end adenylation and linkage of the indexed adapters and enrichment of the fragments by PCR amplification. Libraries were validated considering their fragment size distribution and by their quantification in the Bioanalyzer device (Agilent 2100), using the Agilent DNA 1000 kit according to the manufacturer's protocol. Quantification of each library was then performed by real-time PCR using the KAPA SYBR FAST Universal qPCR kit, according to the manufacturer's protocol, using the StepOnePlus Real-Time PCR System. The cDNA libraries were sequenced on the Illumina HiSeq 1500 System, into a Rapid paired-end flowcell in 200 cycles of 2×101 bp (paired-end technique), according to the standard manufacturer's protocol (Illumina).

## 4.3. Bioinformatics: assembly and annotation of contigs

The generated readings file was extracted using CASAVA-1.8.2 (Illumina), with Q30 quality filter. The assembly was performed using Trinity software (v. 2.1.1) with paired-end parameters [95]. The sample reads were used for assembling again, and the transcripts obtained were used as a reference for the gene expression analyses. Condition-specific expression analysis was performed by aligning the paired-end reads of each of the samples against the assembly of the reference transcriptome, followed by the abundance estimation using the RSEM (RNA-Seq expectation maximization) method [96]. The RSEM aims to estimate the approximate level of expression for each transcript, generating FPKM for each transcript. This method is used due to generation of isoforms, splice variants and duplicate genes in de novo assemblies of transcriptomes, and uses an iterative process to determine reads fractionally for each transcript based on the probability of the reads being derived from each transcript, taking into account the bias in the position generated by the RNA-Seq library creation protocol. The contigs were annotated by Blast script using the Uniprot database. Curation of the annotation was done manually by separating only the contigs corresponding to Hc.

## 4.4. Sequence analysis

The programs provided by the ExPASy Molecular Biology Server of the Swiss Institute of Bioinformatics (http://www.expasy.org) were used for translation and other sequence analyses. N-terminal signal sequences required for export into the extracellular space were predicted with

SignalP 4.1 [97]. Multiple sequence alignment of the amino acid sequences of myriapod and other selected arthropod Hc and PO (electronic supplementary material, table S1) was constructed using the software Clustal X2 [98]. An alignment of the amino acid sequences of Hcs and PPOs as constructed with Clustal X and the BLOSUM 62 matrix. The final alignment covered 62 Hc and 17 PPO sequences from arthropods.

## 4.5. Phylogenetic analysis

We selected the mature protein sequences of the putative Hc without the signal peptide and PO (see electronic supplementary material, table S1) to be used in Prottest 2.4 [99]. Prottest selected the model of protein evolution that best fit in the sequence alignment; an improved general amino acid Neplacement matrix (LG) with site heterogeneity model and gamma plus invariant sites (G + I). The Bayesian analyses were carried out using Markov chain Monte Carlo (MCMC) implemented in BEAST 1.7.5 package [100]. We ran four independent MCMC searches using distinct randomly generated starting trees. Each run consisted of 25 million generations, and trees were sampled every 1000 generations. Convergence was inspected in Tracer v. 1.5 [100]. All runs reached a stationary level after 10% burnin with a large effective sample size. Trees obtained after the burnin step were used to generate a maximum clade credibility tree with TreeAnnotator v. 1.7.5, using a majority rule [100]. The resulting tree was visualized and edited using FigTree v. 1.4.0 (http://tree.bio.ed.ac.uk/software/figtree).

## 4.6. Structural analysis of *Scolopendra subspinipes subspinipes*

The sequence of two subunits of Hc of *Scolopendra subspinipes subspinipes* were aligned to the four sequences from *Scutigera coleoptrata* individually using the method Tcoffee available at Jalview 2.10.4b1 software [101].

Homology modelling of each subunit of the *S. subspinipes subspinipes* Hc sequences was performed using SWISS-MODEL (https://swissmodel.expasy.org/) with Hc subunit II of *Limulus polyphemus* (Protein Data Bank ID: 1OXY) [28] like reference and implemented in PyMOL. The software http://mordred.bioc.cam.ac.uk/~rapper/rampage.php and http://servicesn.mbi.ucla.edu/SAVES/ was used to calculate the Ramachandran and Verified3D, respectively.

The prediction of the secondary structure of the sequences was performed using Promals3D (http://prodata.swmed.edu/promals3d/promals3d.php) [102], the crystal structure of *Panulirus interruptus* Hc (1HCY) and the PO from *Penaeus japonicas* (3WKY) was used as a template too.

## 4.7. Determination of phenoloxidase assay

Dose–response curves using haemolymph samples (haemocytes and plasma) of *Scolopendra subspinipes subspinipes* and *S. viridicornis*, were used to confirm the presence of melanin-synthesis pathway components within an Myriapoda. The activity of PO were determined in triplicate for each of the two samples of each species, using 20 µl of haemolymph (haemocytes or plasma) sample, 40 µl of phosphate buffer (50 mmol l$^{-1}$ pH 7.5), 25 µl of ddH$_2$O, and 30 µl of 10 mmol l$^{-1}$ substrate dopamine hydrochloride (Sigma-

Aldrich H8502) for *o*-diphenoloxidase activity [103]. The sample was incubated for 30 min a 37°C with trypsin or ddH$_2$O before add the substrate dopamine. PPO activity was calculated for each sample and each substrate by repeating the assays using 25 µl of trypsin (0.1 mg ml$^{-1}$ Sigma) instead of ddH$_2$O, to convert PPO to its active and measureable form PO, as per Mydlarz & Palmer [104]. Controls for substrate auto-oxidation were included where the sample extract was replaced with ddH$_2$O. Independent effects of samples were tested (and not detected) in the absence of substrate. Absorbance at 490 nm was recorded every 15 min for 300 min using a Victor$^3$ spectrophotometer (1420 Multilabel Counter/Victor$^3$, Perkin Elmer), and activity determined from the change in absorbance for the linear portion of the reaction curve.

## 4.8. Sample preparation and pH stability for electron transmission microscopy

The haemolymph of *Scolopendra subspinipes subspinipes* was withdrawn without anticoagulant buffer with an apyrogenic syringe. After that, the haemolymph was centrifuged at 3.300$g$ for 30 min at 4°C to separate the haemocytes from plasma. The supernatant (plasma) was submitted to ultracentrifugation at 132.000$g$ a 4°C for 12 h. The Hc pellet was resuspended in TRIS stabilizing buffer (50 mM TRIS, 5 mM CaCl$_2$, 5 mM MgCl$_2$, pH 7,4) through shaking at 4°C for 12 h.

The buffer solutions MES (2-(N-Morpholino) ethanesulfonic acid hydrate, M8250, Sigma) and Tris (Tris(hydroxymethyl) aminomethane, T1503, Sigma) were prepared at dilution on H2Odd. The MES buffer ranges of pH 5.5 at 6.7 and the buffer solution Tris–HCl range of pH 7.0–9.0. The pH was adjusted with NaOH for MES buffer and HCl for Tris buffer. The addition of salts CaCl$_2$ and MgCl$_2$ was made after pH adjusted. After that, the final buffer solution was obtained at 50 mM MES, 5 mM CaCl$_2$ and 5 mM MgCl$_2$ at pH de 6.2 and 6.6; and another buffer 50 mM TRIS–HCl, 5 mM CaCl$_2$ and 5 mM MgCl$_2$ at pH 7.2, 7.5, 7.8, 8.0, 8.2 and 8.5 and they are ready to use.

## 4.9. Transmission electron microscopy

For Hc complex visualization by negative stain, Hc samples of *Scolopendra subspinipes subspinipes* (26–30 µg ml$^{-1}$) at MES buffer (50 mM MES; 5 mM CaCl$_2$; 5 mM MgCl$_2$; pH 6.2 and pH 6.6) and samples at TRIS stabilizing buffer (50 mM TRIS; 5 mM CaCl$_2$; 5 mM MgCl$_2$; pH 7.2–8.5). The images were acquired at a JEOL JEM-1400 Plus 120 kV at a Gatan CCD Multiscan 794 camera (1 k × 1 k pixels). The images collected in each pH condition were counted the number of particles in their quaternary or hexameric state. For this, it was the 'Cell Counter' plugin of Fiji software (ImageJ 1.49 V) [105].

## 4.10. Data collection

The haemolymph of both species (*S. subspinipes subspinipes* and *S. viridicornis*) was withdrawn in the presence of anticoagulant buffer (1 mM Propanol, 154 mM NaCl) with an apyrogenic syringe to avoid haemocyte degranulation. After that, it was centrifuged at 3300$g$ for 30 min at 4°C to separate the haemocytes from plasma. The plasma was purified by size exclusion chromatography using a Superose 6™ 3.2/

royalsocietypublishing.org/journal/rsob  Open Biol. 10: 190258

300 at a linear gradient with TRIS Stabilizing Buffer (50 mM TRIS; 5 mM $CaCl_2$; 5 mM $MgCl_2$, pH 8.0) at flow of 0.1 ml min$^{-1}$. The first peak was used to prepare the negative stain grids. Holey carbon-coated grids were glow discharged for 25 s at 15 mA using an easiGlow system (PELCO). A 3 µl of purified Hc was deposited onto the grid for 60 s, staining twice with 3 µl of 2% uranyl acetate solution (30 s), the solution excess was dried using a filter paper at room temperature and blotting an air-drying. For *S. subspinipes sub-spinipes*, we performed the data acquisition of 184 micrographs in a Talos F200C microscope (Thermo Fisher Scientific, USA), equipped with a Ceta 4 K×4 K camera (Thermo Fisher Scientific, USA), at 200 kV, 57 000 times of magnification, the exposure time was 1 s, with a dose of 17 eÅ$^{-2}$ s$^{-1}$ at −2 µm with a pixel size of 2.58 Å and with a defocus range between −2 µm and −4 µm.

## 4.11. Molecular modelling and visualization

Molecular models of the two different *Scolopendra subspinipes subspinipes* Hc subunits (Ssu1, Ssu2) in oxygenated conformation were built by the SWISS-MODEL software (using the X-ray crystal structure of subunit II of *Limulus polyphemus* (Protein Data Bank ID: 1OXY) [28] like reference and implemented in PyMOL, SWISS-MODEL is freely available at https://swissmodel.expasy.org; the ClustalW software package was applied for the sequence alignment [98,106].

## 4.12. Molecular docking

In order to evaluate the best-binding mode of the chains SSu1 and Ssu2, molecular docking was performed using Frodock server (http://frodock.chaconlab.org). Frodock is a more efficient rigid-body docking tool, in relation to the state-of-the-art rigid-body docking methods [107,108]. Three different combinations were tested: Ssu1 with Ssu1, Ssu1 with Ssu2 and Ssu2 with Ssu2. The results were analysed globally to evaluate the most probable binding sites for each combination. The top 10 scored complexes were analysed according to the expected type of interaction described for *Panulirus interruptus* Hc hexamer. In these analyses, it was also evaluated if the generated complexes were physically possible, considering the steric effect.

## 4.13. Oligomer construction and atomic model fitting

Using the tight dimer obtained in the molecular docking step, we constructed the hexamer. The rigid body fitting was performed for three tight dimers using the Chimera package and the negative staining map. First, each dimer was positioned properly in the map, then the atomic models were fitted into the map allowing rotation and shift parameters. The resulted structures were fitted again using the map simulated from atoms using resolutions ranging from 3.0 Å to 1.5 Å, and the correlation was checked in the end of each attempt. The steps were repeated, varying the shift and rotation parameters, until reach at least 0.6 of correlation between simulated and negative staining maps.

The complete hexamer was triplicated and fitted into the respective densities generating the complete oligomer of the *S. subspinipes subspinipes* Hc, following the same fitting steps of the hexamer.

Data accessibility. This article has no additional data.

Authors' contributions. K.C.T.R. and P.I.S. carried out the molecular laboratory work, sample preparation and drafted the manuscript. A.C.B. carried out buffer stability tests; U.C.O., M.Y.N. and I.L.M.J.-A. performed transcriptome and Bayesian analysis. E.C. participate with the first analyses of transcriptome; A.C.B., A.C., D.C.S. and K.C.T.R. performed the TEM sample preparation and data collection. K.C.T.R. and A.F.-A. contributed with the single particle analyses. K.C.T.R. carried out sequence alignments, K.C.T.R. and J.F.R.M. performed the analysis of docking molecular and rigid body fitting. K.C.T.R., M.v.H., P.I.S. and R.V.P. designed the study, supervised data analysis and revised the manuscript. All authors gave final approval for publication and agree to be held accountable for the work performed therein.

Competing interests. We declare we have no competing interests.

Funding. This work was supported by the Research Support Foundation of the State of São Paulo (FAPESP/CeTICS: grant no. 2013/07467-1 and FAPESP/LNNano: grant no. 2017/15340-2); the Brazilian National Council for Scientific and Technological Development (CNPq) (grant no. 472744/2012-7), Science without Borders during Sandwich PhD (CNPq: grant no. 232252/2014-9), Science without Borders Project (CNPq: grant no. 400796/2012-0); and the Coordination for the Improvement of Higher Education Personne (CAPES-001).

Acknowledgements. We thank Dr Alessandro Giupponi and Dr Rogério Bertani for their unconditional help during the field trips to collect the specimens. Soraia Maria MSc do Nascimento and Thiago de Jesus Oliveira MSc took care of biotherium. We thank Dra Milene Cristina Menezes dos Santos from Proteomics Laboratory of Laboratório de Toxinologia Aplicada—CeTICS/CEPID of Butantan Institute for her enormous help. We thank LNNano/CNPEM for access to the EM facility.

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
