## [Reviewer comments · Open Biology]

Review History

RSOB-19-0258.R0 (Original submission)

Review form: Reviewer 1

Recommendation

Major revision is needed (please make suggestions in comments)

Do you have any ethical concerns with this paper?

No

Comments to the Author

Hemocyanins (Hcs) are oxygen transport proteins exclusively found in certain invertebrates, namely molluscs and arthropods. Within the phylum of arthropods, Hcs are mainly found in chelicerates and crustacean, but also in some hexapoda and myriapoda. Within the latter subphylum, Hcs have been found in diplopoda, and in one case in chilopoda, in the order scutigeroformorpha.

Hcs are grouped together with phenoloxidasases (POs), having a very similar architecture of the respective active sites. POs are part of the innate immune system, and are for example involved in wound healing and sclerotization. In fact, chelicerata do not seem to possess POs, and the respective function is thought to be performed by Hcs, activated e.g. by proteolytic enzymes. The quaternary structure of Hcs of arthropoda comprises multiples of hexamers. Typically, in crustacean 1x6 or 2x6 variants are found, in chelicerates additionally 4x6 or 8x6mers. Characteristic for myriapoda seem to be formation of 3x6 or 6x6 mers, but characterization of the quaternary structure was performed so far only in three cases. Typically, larger quaternary assemblies require a larger number of different subunit types, and go along with a lower affinity, and higher cooperativity.

Arthropod POs are usually found as hexamers in the hemolymph, typically as pro-enzymes. Compared to HCs the amount of POs is very small, while Hc concentration can be as high as 100 mg/ml in certain species.

In the present manuscript, Hcs and POs from another chilopoda, namely two species of the order Scolopendromorpha (*Scolopendra suspinipes* and *Scolopendra viridicornis*) are described. To this end, the sequences were determined, the transcription level addressed and, for the hemocyanin, the protein isolated and the quaternary structure determined by 3D-EM reconstruction.

Overall, the paper seems to be sound. However, the presentation needs to be improved, both with respect to quality of the figures and use of English language (some passages are practically incomprehensible). With respect to this, I only pointed to a few passages. The complete manuscript has to be thoroughly revised in terms of writing.

Specific comments:

Tab. 1 in S1 undecipherable

Page 5: structure of PO: not all POs are hexameric, but the one from crustacean are.

Page 5: The text jumps quite a bit between the different phylogenetic levels. Please be more correct, not every reader is familiar with the myriapod phylogeny.

Fig.1: species undecipherable, please reconsider presentation.

p.9: the discussion about the role of tryptophan needs to be revised, too general and not really clear, what the important factor with respect to Hc/Po function actually is.

Fig.3: "Hemocytos and plasma from both species had protein dispersed...". I don't see anything in case of SviH. I also cannot see six bands in SsuH. Overall, it is difficult to relate the fig. to the text. Please reconsider. Possibly the gels can be omitted.

p.11: The sentences "...activity was observed only in hemocytes after activation ...by trypsin" and "We observed the auto-activation of the hemocytes..." seem to be contradictory in their meaning. Please clarify.

Why is the activity lower in presence of SDS? Usually POs can be activated by SDS....

PO presents a much higher fraction of protein in hemocytes than in plasma, therefore a comparison based on the overall protein concentration is difficult, and therefore one cannot exclude that POs play a role in plasma also. Also reconsider the respective sentences in the discussion (p. 21. Last paragraph).

Fig. 4, legend: what is meant by "dopamine like substrate"?

p.12: the number of particles on the EM grids at the different pH values etc could be better acknowledged if presented as bar chart.

Page 14: SEC profiles: axis title missing. Please specify the SEC column in more detail (supplier, complete name).

Page 14, first paragraph: incomprehensible

Fig.7, legend: A,B,C missing. What is meant by "constricted"?

p. 15: "18 subunits are evenly distributed..." in what respect "evenly" and what is the relevance of that statement?

p.16, Tab.1 (should be 2). Not clear what the different types of hexameric interfaces are, and how the relevant AA were identified.

Fig.10: Isn't the rather large unstructured loop in A on the left side unusual?

p. 19: meaning of "hotspots" unclear. How are they defined? Please describe what the reader should look at in Fig. 12.

Fig.13. The electron density does not have complete three-fold symmetry. Do the fitted hexamers have complete 3-fold symmetry? It is difficult to judge in the fig. Please comment in the text.

Page 21: 4th paragraph: crustacean are also arthropods....please rephrase.

p.22, second last paragraph: typically the concentration of POs in the hemolymph is much lower than Hc. Thus, it seems unlikely that the observed 1x6mers are POs. Please rephrase the corresponding sentence in the abstract.

p.26, structural analysis: why was limulus Hc used for modeling of the subunit, and Panulirus Hc for the hexamer?

p. 28: why were the divalent salts added later? Could that be the reason for some dissociation?

p.29, first paragraph: what is an "open conformation" here?

p.29: Molecular docking: if I understood it correctly, it is not flexible docking, please clarify in the MM

Decision letter (RSOB-19-0258.R0)

02-Jan-2020

Dear Dr da Silva Junior,

We are writing to inform you that the Editor has reached a decision on your manuscript RSOB-19-0258 entitled "Myriapod Hemocyanin: The First 3D Reconstruction of *Scolopendra subspinipes* and Preliminary Structural Analysis of *S. viridicornis*", submitted to Open Biology.

As you will see from the reviewer's comments below, there are a number of criticisms that prevent us from accepting your manuscript at this stage. The reviewers suggest, however, that a revised version could be acceptable, if you are able to address their concerns. If you think that you can deal satisfactorily with the reviewer's suggestions, we would be pleased to consider a revised manuscript.

The revision will be re-reviewed, where possible, by the original referees. As such, please submit the revised version of your manuscript within four weeks. If you do not think you will be able to meet this date please let us know immediately.

When submitting your revised manuscript, please respond to the comments made by the referee(s) and upload a file "Response to Referees" in "Section 6 - File Upload". You can use this to document any changes you make to the original manuscript. In order to expedite the processing of the revised manuscript, please be as specific as possible in your response to the referee(s).

Please see our detailed instructions for revision requirements
<https://royalsociety.org/journals/authors/author-guidelines/>

Sincerely,
The Open Biology Team
mailto: openbiology@royalsociety.org

Reviewer's Comments to Author(s):

Referee:

Comments to the Author(s)

Hemocyanins (Hcs) are oxygen transport proteins exclusively found in certain invertebrates, namely molluscs and arthropods. Within the phylum of arthropods, Hcs are mainly found in chelicerates and crustacean, but also in some hexapoda and myriapoda. Within the latter subphylum, Hcs have been found in diplopoda, and in one case in chilopoda, in the order scutigeroptomorpha.

Hcs are grouped together with phenoloxidasases (POs), having a very similar architecture of the respective actives sites. POs are part of the innate immune system, and are for example involved in wound healing and sclerotization. In fact, chelicerata do not seem to possess POs, and the respective functions is thought to be performed by Hcs, activated e.g. by proteolytic enzymes. The quaternary structure of Hcs of arthropoda comprises multiples of hexamers. Typically, in crustacean 1x6 or 2x6 variants are found, in chelicerates additionally 4x6 or 8x6mers.

Characteristic for myriapoda seem to be formation of 3x6 or 6x6 mers, but characterization of the quaternary structure was performed so far only in three cases. Typically, larger quaternary assemblies require a larger number of different subunit types, and go along with a lower affinity, and higher cooperativity.

Arthropod POs are usually found as hexamers in the hemolymph, typically as pro-enzymes. Compared to HCs the amount of POs is very small, while Hc concentration can be as high as 100 mg/ml in certain species.

In the present manuscript, Hcs and POs from another chilopoda, namely two species of the order Scolopendromorpha (*Scolopendra suspinipes* and *Scolopendra viridicornis*) are described. To this end, the sequences were determined, the transcription level addressed and, for the hemocyanin, the protein isolated and the quaternary structure determined by 3D-EM reconstruction.

Overall, the paper seems to be sound. However, the presentation needs to be improved, both with respect to quality of the figures and use of English language (some passages are practically incomprehensible). With respect to this, I only pointed to a few passages. The complete manuscript has to be thoroughly revised in terms of writing.

Specific comments:

Tab. 1 in S1 undecipherable

Page 5: structure of PO: not all POs are hexameric, but the one from crustacean are.

Page 5: The text jumps quite a bit between the different phylogenetic levels. Please be more correct, not every reader is familiar with the myriapod phylogeny.

Fig.1: species undecipherable, please reconsider presentation.

p.9: the discussion about the role of tryptophan needs to be revised, too general and not really clear, what the important factor with respect to Hc/Po function actually is.

Fig.3: "Hemocytos and plasma from both species had protein dispersed...". I don't see anything in case of SviH. I also cannot see six bands in SsuH. Overall, it is difficult to relate the fig. to the text. Please reconsider. Possibly the gels can be omitted.

p.11: The sentences "...activity was observed only in hemocytes after activation ...by trypsin" and "We observed the auto-activation of the hemocytes..." seem to be contradictory in their meaning. Please clarify.

Why is the activity lower in presence of SDS? Usually POs can be activated by SDS....

PO presents a much higher fraction of protein in hemocytes than in plasma, therefore a comparison based on the overall protein concentration is difficult, and therefore one cannot exclude that POs play a role in plasma also. Also reconsider the respective sentences in the discussion (p. 21. Last paragraph).

Fig. 4, legend: what is meant by “dopamine like substrate”?

p.12: the number of particles on the EM grids at the different pH values etc could be better acknowledged if presented as bar chart.

Page 14: SEC profiles: axis title missing. Please specify the SEC column in more detail (supplier, complete name).

Page 14, first paragraph: incomprehensible

Fig.7, legend: A,B,C missing. What is meant by “constricted”?

p. 15: “18 subunits are evenly distributed...” in what respect “evenly” and what is the relevance of that statement?

p.16, Tab.1 (should be 2). Not clear what the different types of hexameric interfaces are, and how the relevant AA were identified.

Fig.10: Isn't the rather large unstructured loop in A on the left side unusual?

p. 19: meaning of “hotspots” unclear. How are they defined? Please describe what the reader should look at in Fig. 12.

Fig.13. The electron density does not have complete three-fold symmetry. Do the fitted hexamers have complete 3-fold symmetry? It is difficult to judge in the fig. Please comment in the text.

Page 21: 4th paragraph: crustacean are also arthropods....please rephrase.

p.22, second last paragraph: typically the concentration of POs in the hemolymph is much lower than Hc. Thus, it seems unlikely that the observed 1x6mers are POs. Please rephrase the corresponding sentence in the abstract.

p.26, structural analysis: why was limulus Hc used for modeling of the subunit, and Panulirus Hc for the hexamer?

p. 28: why were the divalent salts added later? Could that be the reason for some dissociation?

p.29, first paragraph: what is an “open conformation” here?

p.29: Molecular docking: if I understood it correctly, it is not flexible docking, please clarify in the MM

Author's Response to Decision Letter for (RSOB-19-0258.R0)

See Appendix A.

RSOB-19-0258.R1 (Revision)

Review form: Reviewer 1

Recommendation

Accept with minor revision (please list in comments)

Do you have any ethical concerns with this paper?

No

Comments to the Author

The authors addressed all of my concerns appropriately.

There is only one aspect I suggest to change:

if the 2D-Gel are not shown, they should not be discussed, and the corresponding part in the methods-part should be deleted.

My problem was, that I cannot see what the authors describe, like six bands in case of Ssuh. This problem does not disappear by not showing the gels. So I suggest to completely deleted all parts referring to the gels. That does not change the story.

Decision letter (RSOB-19-0258.R1)

02-Mar-2020

Dear Dr da Silva Junior,

We are pleased to inform you that your manuscript RSOB-19-0258.R1 entitled "Myriapod Hemocyanin: The First 3D Reconstruction of *Scolopendra subspinipes* and Preliminary Structural Analysis of *S. viridicornis*" has been accepted by the Editor for publication in Open Biology. The reviewer(s) have recommended publication, but also suggest some minor revisions to your manuscript. Therefore, we invite you to respond to the reviewer(s)' comments and revise your manuscript.

Please submit the revised version of your manuscript within 7 days. If you do not think you will be able to meet this date please let us know immediately and we can extend this deadline for you.

- 1) A text file of the manuscript (doc, txt, rtf or tex), including the references, tables (including captions) and figure captions. Please remove any tracked changes from the text before submission. PDF files are not an accepted format for the "Main Document".
- 2) A separate electronic file of each figure (tiff, EPS or print-quality PDF preferred). The format should be produced directly from original creation package, or original software format. Please note that PowerPoint files are not accepted.
- 3) Electronic supplementary material: this should be contained in a separate file from the main text and meet our ESM criteria (see <http://royalsocietypublishing.org/instructions-authors#question5>). All supplementary materials accompanying an accepted article will be treated as in their final form. They will be published alongside the paper on the journal website and posted on the online figshare repository. Files on figshare will be made available

approximately one week before the accompanying article so that the supplementary material can be attributed a unique DOI.

Online supplementary material will also carry the title and description provided during submission, so please ensure these are accurate and informative. Note that the Royal Society will not edit or typeset supplementary material and it will be hosted as provided. Please ensure that the supplementary material includes the paper details (authors, title, journal name, article DOI). Your article DOI will be 10.1098/rsob.2016[last 4 digits of e.g. 10.1098/rsob.20160049].

4) A media summary: a short non-technical summary (up to 100 words) of the key findings/importance of your manuscript. Please try to write in simple English, avoid jargon, explain the importance of the topic, outline the main implications and describe why this topic is newsworthy.

Images

Data-Sharing

It is a condition of publication that data supporting your paper are made available. Data should be made available either in the electronic supplementary material or through an appropriate repository. Details of how to access data should be included in your paper. Please see <http://royalsocietypublishing.org/site/authors/policy.xhtml#question6> for more details.

Data accessibility section

Sincerely,

The Open Biology Team

<mailto:openbiology@royalsociety.org>

Reviewer(s)' Comments to Author:

Referee: 1

Comments to the Author(s)

The authors addressed all of my concerns appropriately.

There is only one aspect I suggest to change:

if the 2D-Gel are not shown, they should not be discussed, and the corresponding part in the methods-part should be deleted.

My problem was, that I cannot see what the authors describe, like six bands in case of Ssuh. This problem does not disappear by not showing the gels. So I suggest to completely deleted all parts referring to the gels. That does not change the story.

Author's Response to Decision Letter for (RSOB-190258.R0)

See Appendix B.

Decision letter (RSOB-19-0258.R2)

06-Mar-2020

Dear Dr da Silva Junior,

We are pleased to inform you that your manuscript entitled "Myriapod Hemocyanin: The First 3D Reconstruction of *Scolopendra subspinipes* and Preliminary Structural Analysis of *S. viridicornis*" has been accepted by the Editor for publication in Open Biology.

Article processing charge

Please note that the article processing charge is immediately payable. A separate email will be sent out shortly to confirm the charge due. The preferred payment method is by credit card; however, other payment options are available.

Sincerely,

The Open Biology Team

mailto: openbiology@royalsociety.org

Appendix A

Reviewer's Comments to Author(s):

Referee:

Comments to the Author(s)

Hemocyanins (Hcs) are oxygen transport proteins exclusively found in certain invertebrates, namely molluscs and arthropods. Within the phylum of arthropods, Hcs are mainly found in chelicerates and crustacean, but also in some hexapoda and myriapoda. Within the latter subphylum, Hcs have been found in diplopoda, and in one case in chilopoda, in the order scutigermorpha.

Hcs are grouped together with phenoloxidasases (POs), having a very similar architecture of the respective actives sites. POs are part of the innate immune system, and are for example involved in wound healing and sclerotization. In fact, chelicerata do not seem to possess POs, and the respective functions is thought to be performed by Hcs, activated e.g. by proteolytic enzymes.

The quaternary structure of Hcs of arthropoda comprises multiples of hexamers. Typically, in crustacean 1x6 or 2x6 variants are found, in chelicerates additionally 4x6 or 8x6mers. Characteristic for myriapoda seem to be formation of 3x6 or 6x6 mers, but characterization of the quaternary structure was performed so far only in three cases. Typically, larger quaternary assemblies require a larger number of different subunit types, and go along with a lower affinity, and higher cooperativity.

Arthropod POs are usually found as hexamers in the hemolymph, typically as pro-enzymes. Compared to HCs the amount of POs is very small, while Hc concentration can be as high as 100 mg/ml in certain species.

In the present manuscript, Hcs and POs from another chilopoda, namely two species of the order Scolopendromorpha (*Scolopendra suspinipes* and *Scolopendra viridicornis*) are described. To this end, the sequences were determined, the transcription level addressed and, for the hemocyanin, the protein isolated and the quaternary structure determined by 3D-EM reconstruction.

Overall, the paper seems to be sound. However, the presentation needs to be improved, both with respect to quality of the figures and use of English language (some passages are practically uncomprehensible). With respect to this, I only pointed to a few passages. The complete manuscript has to be thoroughly revised in terms of writing.

Specific comments:

Tab. 1 in S1 undecipherable

Answer: I Changed the figure

Page 5: structure of PO: not all POs are hexameric, but the one from crustacean are.

Answer: I put it on text

The geometry and coordination environment of the active site of the arthropod PO are very similar to those of the arthropod Hc; and while not all POs are hexameric [33], those from crustaceans are [34,35].

Arthropod Hc proteins have been best studied in chelicerates and crustaceans [32,36–43]. Hc has also been found in Hexapoda [44–47] and Onycophora [48], but these subphyla, including Myriapoda, have long been ignored in this regard, because of the mediation of their O₂ supply via their trachea [49].

Page 5: The text jumps quite a bit between the different phylogenetic levels. Please be more correct, not every reader is familiar with the myriapod phylogeny.

Answer: I put it on text.

Myriapoda is a group of arthropods divided into four classes, namely Chilopoda (centipedes), Diplopoda (millipedes), Symphyla and Pauropoda [1].

Millipedes are a mega-diverse group of arthropods and are among the most important consumers of detritus in many terrestrial ecosystems. Comprising more than 12,000 described species[2], millipedes are found on six continents and in virtually all of Earth's biomes [3]. This group is characterized by the segments in which two pairs of legs are arranged on one body segment [4].

The remaining two classes, Pauropoda and Symphyla, are small, translucent, soil-dwelling myriapods, with body lengths of less than 2 mm and 1-8 mm, respectively. The symphylids have long and filiform antennae and a pair of specialized appendages at the preanal segment, called spinnerets, while the pauropods have distinctive antennae, which are branching and have long flagella. A total of 835 pauropods species in 2 orders and 5 families and 195 symphylid species in one order and 2 families have been described to date [5,6].

Centipedes (Chilopoda), one of the four major lineages of Myriapods (Arthropoda), constitute the only predatory myriapod group. They live in many terrestrial habitats [7] with fossil records spanning 420 million years. A key trait of this group is a pair of poisonous claws formed from a modified first appendage [4]. They comprise approximately 3,300-3,500 species distributed in all continents except Antarctica [8] with the greatest diversity occurring in the tropics and warm temperate zones. Most species inhabit leaf litter and soil or are found under stones, bark, or wood in forests, although grassland, desert, caves, and the coastal areas are also occupied by some species [9].

The internal organs of all centipedes, except Scutigromorpha, are supplied with oxygen through trachea, spirally thickened chitinous tubules of ectodermal origin, and which originate from laterally placed openings, the spiracles [10]. Some species have occludable spiracles, a prerequisite for discontinuous respiration and their tracheal ultrastructure is similar

to that in insects and arachnids. Other species have tracheal lungs with short tracheal tufts, while still others have an insect-like tracheal system [11].

The tracheal and circulatory systems in Myriapoda are well developed. The circulatory system of chilopods is of the open kind [12]. Analogous to blood, the fluid that circulates in arthropods is called hemolymph. It is pumped by the heart into a body cavity called a hemocoel, where it sloshes around and bathes the internal organs in nutrients and gases [13]. The hemolymph of many arthropods and mollusks presents a protein called hemocyanin, a large copper-containing protein that transports and store oxygen [14,15]. Hemocyanin is not found in blood cells but is found freely dissolved in the hemolymph. It forms the major protein constituent (50% to 90%) of this fluid, with concentrations of up to 120 mg/mL [16]. Besides hemocyanin, prophenoloxidase (PPO) is also the hemocytes as an inactive proenzyme [17]. These proteins are important for primary immune responses, with hemocyanin fragments in plasma, acting like antimicrobial peptides (PvHCt [18] and rondonin [19]) and phenoloxidase (PO) enzymes acting as wound healer and playing a role in sclerotization and melanization [20–22].

Fig.1: species undecipherable, please reconsider presentation.

Answer: I changed the figure

p.9: the discussion about the role of tryptophan needs to be revised, too general and not really clear, what the important factor with respect to Hc/Po function actually is.

Answer: I rewrote this part.

Phenoloxidase (PO; EC 1.14.18.1) and tyrosinase, members of the family of type-3 copper proteins, each catalyzes the hydroxylation of monophenol compounds to o-diphenol (monophenoloxidase activity) and subsequent oxidation to produce the corresponding o-quinone (o-diphenoloxidase activity). In contrast, the type 3 dicopper-site-containing members of the protein family called catechol oxidase catalyze only the latter diphenoloxidase reaction [62,63].

Our inspection of the multiple sequence and structure alignment with seven subunits of spider *Aphonopelma hentzi* (EcaHcA-G), the crystal structure of hemocyanin from the crustacean *Panulirus interruptus* (1HCY) and the crystal structure of PO of *Penaeus japonicus* (3WKY) and of PO of the insect *Manduca sexta* (3HHSA-B) indicated, the presence of tryptophane near the first histidine (H1-CuA) in all sequences of PO (3WKY, 3HHSA-B) and sequences of hemocyanin with PO activity (EcaHcA-G); in contrast, our inspection of the sequence/structure of hemocyanin with only oxygen transport activity showed a valine in the

crustacean protein (1HCY) and tyrosine in the myriapods ones (Fig – 02; S1Figure) instead of the tryptophan [34,64] .

A conserved Phe (F) residue called “place holder” occurs in the active site pocket before activation and is thought to block access to the substrate is present in all type 3 copper protein. When PPO is activated, this place holder must be removed from the active site pocket. When the blocking residue (F⁸⁴) in DmPPO3 (*Drosophila melanogaster*) was mutated into tryptophan (W), which has a hydrophobic side chain, DmPPO3(W⁸⁴) activity significantly decreased after being activated by ethanol [65].

Tryptophan contains a non-carbon atom (nitrogen) in the aromatic ring, making this residue more reactive than phenylalanine although less so than than tyrosine. Tryptophan can play a role in binding to non-protein atoms while the tyrosine side chain, being partially hydrophobic, prefers to be buried in the hydrophobic core. Also, the aromatic tyrosine side chain can be involved in π - π stacking interactions with other aromatic side chains. The valine side chain is small but completely aliphatic and hydrophobic and hence non-reactive, and is thus rarely directly involved in protein functions like catalysis, although it can play a role in substrate recognition. The hydrophobic aromatic amino acid residues can sometimes substitute for aliphatic residues of a similar size, for example phenylalanine for leucine, but not the large tryptophan for the small valine. In particular, hydrophobic amino acid residues can be involved in binding/recognition of hydrophobic ligands such as lipids [66].

Although the di-copper active sites of all of the above-mentioned type 3 copper proteins could be well superimposed, interesting differences have been observed, mainly at the CuA site [33]. In particular, tyrosine-switched tryptophan may have made the region less hydrophobic by preventing the entry of substrate, which may explain the probable loss of PO activity from the two hemocyanin proteins identified in *Scolopendra subspinipes subspinipes* (**Error! Reference source not found.**).

Fig.3: “Hemocytes and plasma from both species had protein dispersed...”. I don’t see anything in case of SviH. I also cannot see six bands in SsuH. Overall, it is difficult to relate the fig. to the text. Please reconsider. Possibly the gels can be omitted.

Answer: I change this in text and omitted the gel.

Hemolymphs (lysates of hemocytes and plasma, separately) from *Scolopendra subspinipes subspinipes* and *S. viridicornis* samples were analyzed using 2D-PAGE (data not shown). The lysate of hemocytes from *S. subspinipes* showed six bands in the region between 98 kDa and 62 kDa, each at a different isoelectric point, and two bands were observed in the

gel of the lysate of the plasma. These gel results together were consistent with the six sequences of phenoloxidase and two of hemocyanin identified in the transcriptome of *S. subspinipes subspinipes*. In the gel of the lysate of the plasma from *S. viridicornis*, two bands were observed in the same region as observed for *S. subspinipes*, suggesting the presence of two subunits of hemocyanin here as well. No bands were observed in the gel of the lysate of the hemocytes from *S. viridicornis*.

p.11: The sentences “..activity was observed only in hemocytes after activation ...by trypsin” and “We observed the auto-activation of the hemocytes...” seem to be contradictory in their meaning. Please clarify.

Answer: I changed this phrase

In this study, we used a concentration of 20 µg/mL from lysate of hemocytes and plasma for both species to verify the *o*-diphenoloxidase incubating with dopamine. The activation of prophenoloxidase to phenoloxidase by trypsin (square gray) and the activity of the lysate of hemocytes with buffer (triangle blue) occurred for both species (Figure 1A, B), suggesting that serine protease, the enzyme responsible for this activity, was also present in the hemocytes. Previous studies showed an activation of phenoloxidase activity for myriapods by zymosan and chymotrypsin [76], corroborating our results.

Why is the activity lower in presence of SDS? Usually POs can be activated by SDS.... PO presents a much higher fraction of protein in hemocytes than in plasma, therefore a comparison based of the overall protein concentration is difficult, and therefore one cannot exclude that POs play a role in plasma also. Also reconsider the respective sentences in the discussion (p. 21. Last paragraph).

Answer: Previous studies showed an activation of phenoloxidase activity for myriapods by zymosan and chymotrypsin (Xylander WER. 1992 Immune Defense Reactions of Myriapoda — A Brief Presentation of Recent Results. *8th Int. Congr. Myriapodology*), corroborating our results. Probably the phenoloxidase of plasma occurs after the hemocytes degranulation. In our experiments, we took care to avoid degranulation that is responsible to liberate the granules with PO in plasma. Thus, in this case, we can conclude that PO activity is only in hemocytes lysates. The mechanism of activation of PO in arthropods are still in study and not clean.

Fig. 4, legend: what is meant by “dopamine like substrate”?

Answer: I used the dopamine like substrate for phenoloxidase activity. The formation of dopaminechrome is possible to measure using a spectrophotometer.

Figure 1 – Phenoloxidase activity for both species of *Scolopendra*. A – *Scolopendra subspinipes subspinipes* (SsuH - hemocytes - left; SsuP - plasma - right); B – *Scolopendra viridicornis* (SviH - hemocytes - left; SviP -plasma - right). The activity was measure Victor³

spectrophotometer (1420 Multilabel Counter/Victor³, Perkin Elmer). The samples (hemocyte (SsuH and SviH) or plasma (SsuP and SviP)) were incubated with dopamine (substrate for PPO) in Phosphate Buffer with 30 minutes at 37°C and after that activated with ultrapure water (blue), SDS (orange) or tripsine (gray). The increase of absorbance reveals the phenoloxidase activity by the formation of dopaminechrome.

p.12: the number of particles on the EM grids at the different pH values etc could be better acknowledged if presented as bar chart.

Answer: I did it.

Figure 2 – Number of particles and ratio between the hexamer and complex at the range of pH (6.2 – 8.50). A- The number of particles was counted using the program "Cell Counter" plugin of Fiji software (ImageJ 1.49V). The 1x6-mer was represent by light grey and the complex 3x6-mer by grey. B – The ratio between 1x6-mer and 3x6-mer were calculate.

Page 14: SEC profiles: axis title missing. Please specify the SEC column in more detail (supplier, complete name).

Answer: I corrected it

Chromatogram profile of Plasma. The plasma was separated from hemolymph by centrifugation at 3.300 xG, 30 min at 4°C and the supernatant was submitted to purifier AKTA with column Superose 6 3.2/300 (Ge Healthcare) with isocratic buffer (Tris Buffer pH 8.2) with flow 0.1ml/min. A – *Scolopendra subspinipes subspinipes*; B – *S. viridicornis*

Page 14, first paragraph: incomprehensible

Answer: The paragraph was rewrote.

The hemocyanins of the animals were purified using a Superose 6™ 3.2/300 (Ge Healthcare) SEC column on an FPLC (AKTA) with TRIS as stabilizing buffer at a flow of 0.1 mL/min (**Error! Reference source not found.**). We found two major peaks in both chromatographic profiles. The first peak corresponded to a high level of oligomerization for

the complex. We were initially expecting to find a 6 x 6-mer-type complex of *Scolopendra subspinipes subspinipes* hemocyanin, as was found for hemocyanin from the centipede *Scutigera coleoptrata*, an organism also from the Chilopoda class [54] — but the retention volume of our chromatographic profile (**Error! Reference source not found.**A) indicated only the 3 x 6-mer-type complex. This chromatographic result was corroborated by an analysis of transmission electron microscopy images of negatively stained samples, which showed a 3 x 6-mer-type complex structure, specifically with the three hexamers related to each other by a symmetry rotation axis of 120° and showing a diameter of approximately 29 nm (**Error! Reference source not found.**).

Fig.7, legend: A,B,C missing. What is meant by “constricted”?

Answer: I put A,B, C at the figure and constricted was a error, the correct form is stained

Hemocyanin samples stained with 2% uranyl acetate.

p. 15: “18 subunits are evenly distributed...” in what respect “evenly” and what is the relevance of that statement?

Answer: The subunits are distributed equally in the three hexamers that constituted the protein. The 3x6-mer structure with D3 symmetry was also observed for the hemocyanin from the diplopod *Polydesmus angustus*. Densitometry results showed the presence of the three subunit types in equimolar proportions, with the 18 subunits forming the complex being evenly distributed among two zones: a core zone with six subunits (PanHc1) and a mantle zone with 12 subunits (PanHc2 and PanHc3) [57].

p.16, Tab.1 (should be 2). Not clear what the different types of hexameric interfaces are, and how the relevant AA were identified.

Answer: I wrote some paragraphs to explain this and I changed the number of table.

Since specific interactions have been shown to be important to the assembly of the final 6 x 6-mer structure of *Scutigera coleoptrata* hemocyanin, we performed an alignment of its sequence with our two sequences, i.e., of Ssu1 and Ssu2 (**Error! Reference source not found.**).

Observations of the mantle zone indicated a lack of any contact between two subunits from each hexamer and any other hexamer — and indicated the remaining two subunits to be attached to their counterpart in the adjacent hexamer, together forming the three peripheral bridges. In the 6 x 6-mer assembly of *S. coleoptrata* hemocyanin, this type of bridge has been

designated as a 3↔4 interface [54,57]. This interface in Ssu1 formed from residues 402HYKLTYP A409, 633TH634 and in Ssu 2 from 361VEQLTWP D368, 587TH588.

In the *S. coleoprata* 6 x 6-mer structure, the two 3 x 6-mer half structures have been previously noted to be connected in the mantle by a type of inter-hexamer bridge that has been termed a 3↔3 interface [54]. We observed the presence of histidine (147NRGEHYDRIPi156, 442TKH444) in the ScoHcC subunit; histidine is an important amino acid able to participate in π - π stacking interactions and it was responsible for stabilizing the structure. Our inspection of the Ssu1 sequence at 141IRGNQ-NPVVNL154 and 432TRQ434 showed a gap in this region and the absence of an amino acid residue able to perform the same interaction. And our inspection of Ssu2 at 106QKALRNDKDIVVD118 and 401TGP403 showed an arginine in the place of the histidine; the arginine here may support allostery rather than blocking potential dimerization of 3 x 6-mers [54].

The second type of bridge connecting the 3 x 6-mers in the core of *S. coleoprata* is termed the 1A↔1E interface, with this interface established by interactions of basic residues (KAKK) in the β 3D \rightarrow β 3E loop with acidic residues (ED) at the end of helix α 3.5. Comparable charges (ENKK and PSD) were observed in this region of Ssu2 (Fig. S2).

The three central inter-hexamer bridges of the 3 x 6-mer have been observed to be formed by the core subunit Ssu2, and this type of contact has been termed a 1↔2 interface [54]. Each of the three bridges was formed by a pair of such interfaces together being made up of a large number of residues (**Error! Reference source not found.**). In *S. coleoprata* hemocyanin, each 1↔2 interface pair included a histidine cluster that may support allostery [54].

Fig.10: Isn't the rather large unstructured loop in A on the left side unusual?

Answer: A multiple sequence and structure alignment with 1OXY showed a gap (of 20 amino acid residues) in the N-terminal region of Ssu1 (Figure S4), which explained the loop created at this region (**Error! Reference source not found.A**) when compared with Ssu2 (**Error! Reference source not found.B**). In *Polydemos angustus* hemocyanin, an unusually large histidine-rich loop of 30 amino acid residues (11 histidines) was observed to be located between β -strands 3A and 3B of the subunit PanHc1 [57].

p. 19: meaning of “hotspots” unclear. How are they defined? Please describe what the reader should look at in Fig. 12.

Answer: The tool considers one molecule rigid while make translational changes to the other in order to dock it. The hot spots are the translational center of the highly scored poses, which are being docked into the rigid molecule. They are represented by spheres.

It was added to the text: Additionally, the molecular docking results indicated a more dispersive pattern of hotspots for the Ssu2/Ssu2 solutions than for Ssu1/Ssu1 and Ssu1/Ssu2 (**Error! Reference source not found.A–C**). The top 20% of the docking solution of Ssu1/Ssu1, Ssu1/Ssu2, and Ssu2/Ssu2 are represented by colored spheres, with dark blue representing a poor score and red a good score, in **Error! Reference source not found.D–F**. This pattern demonstrated the potential of an Ssu2 chain to interact with another Ssu2 chain in several different ways, or that Ssu2 could be interacting with more than one chain simultaneously. Assuming the last hypothesis, the Ssu2 chain would be localized in the middle (core zone) of the 3 x 6 hexamer and interacting with other Ssu2 chains inside the hexamer (1↔2 interface) and hence could provide inter-hexamer interactions.

Fig. 12. Molecular docking hotspots from dark blue (poor score) to red (good score) was made using Frodock server. (A) 100% of the Ssu1/Ssu1 docking solutions, (B) 100% of the Ssu1/Ssu2 docking solutions, (C) 100% of the Ssu2/Ssu2 docking solutions, (D) the best 20% of the Ssu1/Ssu1 docking solutions, (E) the best 20% of the Ssu1/Ssu2 docking solutions, (F) the best 20% of the Ssu2/Ssu2 docking solutions. A more dispersive pattern is present in Ssu2/Ssu2 docking solutions, which can be related to the Ssu2 chain being involved in interactions with other chains providing the inter-hexamer interaction.

Fig.13. The electron density does not have complete three-fold symmetry. Do the fitted hexamers have complete 3-fold symmetry? It is difficult to judge in the fig. Please comment in the text.

Answer. I added this phrase on text.

Based on symmetry considerations previously determined for the *S. coleoptrata* 6 x 6-mer, and applied to the 3 x 6-mer found in *S. subspinipes subspinipes*, we suggested that the constituent 1x6-mers are identical with respect to their subunit composition [54,57]

Page 21: 4th paragraph: crustacean are also arthropods...please rephrase.

Answer: I changed the phrase

In the kuruma prawn *Marsupenaeus japonicus*, prophenoloxidase and Hc are synthesized in the hepatopancreas with an N-terminal signal peptide [87], and are secreted to plasma [35], whereas the other well-known arthropod phenoloxidase is synthesized in hemocyte cells without the signal peptide [88,89]. Moreover the quaternary structure of crustacean PO forms a hexamer, similar to that of hemocyanin, whereas the insect PO usually forms a dimer [33,90,91].

p.22, second last paragraph: typically the concentration of POs in the hemolymph is much lower than Hc. Thus, it seems unlikely that the observed 1x6mers are POs. Please rephrase the corresponding sentence in the abstract.

Answer: I changed the phrase.

While we found a relatively small amount of the 3 x 6-mer hemocyanin form in *Scolopendra subspinipes subspinipes* and found the 6 x 6-mer form in *Scolopendra viridicornis*, phenoloxidases were also found to be present in the hemolymph. Note in this regard that the 1 x 6-mer quaternary organization has also been shown for hexamers of PPO [34]. More studies of hemocyanin of *Scolopendra viridicornis* will be necessary to explain the presence of 6 x 6-mer hemocyanin.

p.26, structural analysis: why was limulus Hc used for modeling of the subunit, and Panulirus Hc for the hexamer?

Answer: We used *Limulus polyphemus* to create a subunit in oxygenated form, this is often used to create almost hemocyanin subunits once this is the first crystal structure published and the structure of *Panulirus interruptus* was used to check the interfaces responsible to create a hexamer. I rewrite this part of methods.

Homology modeling of each subunit of the *S. subspinipes subspinipes* hemocyanin sequences was performed using Swiss Model (<https://swissmodel.expasy.org/>) with hemocyanin subunit II of *Limulus polyphemus* (Protein Data Bank ID: 1OXY) [28] and implemented in PyMOL. The software <http://mordred.bioc.cam.ac.uk/~rapper/rampage.php> and <http://servicesn.mbi.ucla.edu/SAVES/> was used to calculate the Ramachandran and Verified3D, respectively.

p. 28: why were the divalent salts added later? Could that be the reason for some dissociation?

Answer: First we prepare the solution MES and TRIS, after that the pH was adjusted and so the salts added.

The buffer solution MES (2-(N-Morpholino) ethanesulfonic acid hydrate, M8250, Sigma) e Tris (Tris(hydroxymethyl) aminomethane, T1503, Sigma) were prepared at dilution on H₂O. The MES buffer ranges of pH 5.5 at 6.7 and the buffer solution Tris-HCl range of pH 7.0 to 9.0. The pH was adjusted with NaOH for MES buffer and HCl for Tris buffer. The addition of salts CaCl₂ and MgCl₂ was made after pH adjusted. After that, the final buffer solution was obtained at 50mM MES, 5mM CaCl₂ and 5mM MgCl₂ at pH de 6.2 and 6.6; and another buffer 50mM Tris-HCl, 5mM CaCl₂ and 5mM MgCl₂ at pH 7.2, 7.5, 7.8, 8.0, 8.2 and 8.5 and they are ready to use.

p.29, first paragraph: what is an “open conformation” here?

Answer: I changed the phrase. Molecular models of the two different *Scolopendra subspinipes subspinipes* hemocyanin subunits (Ssu1, Ssu2) in oxygenated conformation were built by the Swiss Model Software (using the X-ray crystal structure of subunit II of *Limulus polyphemus* (Protein Data Bank ID: 1OXY) [28] like reference and implemented in PyMOL, SWISS-MODEL is freely available at <https://swissmodel.expasy.org>; the ClustalW software package was applied for the sequence alignment [100,108].

p.29: Molecular docking: if I understood it correctly, it is not flexible docking, please clarify in the MM

Answer: Frodock is a more efficient rigid-body docking tool, in relation to the state-of-art rigid-body docking methods according to GARZON *et.al.* 2009.

Ref. GARZON, José Ignacio et al. FRODOCK: a new approach for fast rotational protein–protein docking. **Bioinformatics**, v. 25, n. 19, p. 2544-2551, 2009.

Appendix B

Reviewer(s)' Comments to Author:

Referee: 1

Comments to the Author(s)

The authors addressed all of my concerns appropriately.

There is only one aspect I suggest to change:

if the 2D-Gel are not shown, they should not be discussed, and the corresponding part in the methods-part should be deleted.

My problem was, that I cannot see what the authors describe, like six bands in case of Ssuh. This problem does not disappear by not showing the gels. So I suggest to completely deleted all parts referring to the gels. That does not change the story.

Answer: We did what was solicited by referee.